# InsNet-CRAFTY v1.0: Integrating institutional network dynamics powered by large language models with land use change simulation

Yongchao Zeng[1], Calum Brown [1,2], Mohamed Byari[1☆],Joanna Raymond[1☆],Thomas Schmitt[1☆], Mark Rounsevell[1,3,4]

[1] Institute of Meteorology and Climate Research, Atmospheric Environmental Research (IMK-IFU), Karlsruhe Institute of Technology, 82467 Garmisch-Partenkirchen, Germany

[2] Highlands Rewilding Limited, The Old School House, Bunloit, Drumnadrochit IV63 6XG, UK

[3] Institute of Geography and Geo-ecology, Karlsruhe Institute of Technology, 76131 Karlsruhe, Germany

[4] School of Geosciences, University of Edinburgh, Drummond Street, Edinburgh EH8 9XP, UK

☆  These authors contributed equally to this work.

**Correspondence:** Yongchao Zeng (yongchao.zeng@kit.edu)

**Abstract:** To foster sustainable land use and management, it is crucial – but challenging – to enhance our understanding of how policy interventions influence decision-making actors, and how these interactions can be effectively modelled. Key challenges include endowing modelled actors with autonomy, accurately representing their relational network structures, and managing the often-unstructured information exchange among them. Large language models (LLMs) offer new ways to address these challenges through the development of agents that are capable of mimicking reasoning, reflection, planning, and action. We present InsNet-CRAFTY (Institutional Network – Competition for Resources between Agent Functional Types) v1.0, a multi-LLM-agent model with a polycentric institutional framework coupled with an agent-based land system model. The institutional framework includes a high-level policy-making institution, two lobbyist organisations, two operational institutions, and two advisory agents. For exploratory purposes, illustrative numerical experiments simulating two competing policy priorities are conducted: increasing meat production versus expanding protected areas for nature conservation. We find that the high-level institution tends to avoid radical changes in budget allocations and adopts incremental policy goals for the operational institutions, but it leaves an unresolved budget deficit in one institution and a surplus in another. This is due to the competing influence of multiple stakeholders, which leads to the emergence of a path-dependent decision-making approach. Despite errors in information and behaviours by the LLM agents, the network maintains overall behavioural believability. The results highlight both the potential and the risks of using LLM agents to simulate policy decision-making. While LLM agents demonstrate high flexibility and autonomy in modelling human decision-making and institutional dynamics, their integration with existing land use models is complex, requiring careful workflow design to ensure reliability. These insights contribute to advancing land system modelling and the broader field of institutional analysis, providing new tools and methodologies for researchers and policy-makers.

# 1. Introduction

Scientists have developed various models to study key topics in the land system, such as climate mitigation pathways (Duffy et al., 2022), carbon storage (Ekholm et al., 2024), human fire use (Perkins et al., 2024), and land cover change (Calvin et al., 2022; Chen et al., 2019). Land systems encompass both natural and human factors, with policy interventions playing a pivotal role in shaping land use dynamics (Paz et al., 2020; Wang et al., 2018; Zeng et al., 2025b). These interventions serve as critical mechanisms for addressing climate change, preserving biodiversity, and ensuring food security (Broussard et al., 2023; Guo et al., 2024; Qi et al., 2018). The formation and implementation of land use policies are the product of complex institutional dynamics including the interactions, adaptations, and power relations of the governing bodies involved in policy-making over time. Institutional dynamics can involve a wide range of actors with differing objectives and powers (Davidson et al., 2024), as well as multi-level governance systems, such as that of the European Union (EU) (González, 2016).

In the context of the EU, supranational and national governments' policy implementation can vary based on economic, social and environmental priorities. Local actors, including farmers and regional governments, further influence land use through on-the-ground practices and lobbying efforts. These interactions, whether cooperative or contentious, can result in policy outcomes that either advance or hinder environmental goals. For instance, tensions between biodiversity conservation objectives and agricultural production have led to debates over subsidies and land management practices (Henle et al., 2008; Mattison and Norris, 2005). Understanding how these actors interact and public policies evolve is crucial for assessing how changes in policy can influence the land system in the future, and what this can mean for environmental goals.

Despite the importance of being able to simulate the effects of institutional dynamics on land systems, and despite ample empirical evidence highlighting interconnectivity among institutional actors (Ariti et al., 2019; Díez-Echavarría et al., 2023; Tesfaye et al., 2024), there is a scarcity of land use models which incorporate institutional networks (i.e., networks composed of interacting institutional actors), due to the challenges of representing heterogeneous, autonomous institutional decision-makers. Among the few studies that have explicitly modelled institutional actors within the land system are González (2016) and Holzhauer et al. (2019). In these examples, institutional actors are modelled as rule-based and programmed computational decision-making entities (i.e., agents) that take limited actions in response to specific land use changes. To strengthen the connection between modelling and real-world policy-makers, Zeng et al. (2025b) developed an endogenous institutional model using a fuzzy logic controller mechanism that can integrate real-world policy-makers' knowledge as IF-THEN rules. Other studies employ the network of action situation (NAS) approach (Kimmich et al., 2023), which is developed from action situation and game theory (McGinnis, 2011), allowing for systematic integration of a wide range of empirical evidence. However, NAS is still in its infancy (Tan et al., 2023), and it does not yet offer an approach to create formalized models.

Conventional rule-based or hard-coded models have advantages in modelling specific aspects of policy institutions. However, advancing the holistic representation of institutional actors in formal models needs to overcome three key challenges: agent autonomy, complex relational structures, and unstructured data. Firstly, modelling institutional actors' autonomy requires accounting for heterogeneous behaviour (Dakin and Ryder, 2020), involving learning and memory (Nair and

Howlett, 2017) together with bounded rationality (Jones, 2003; Simon, 1972). Secondly, there are both horizontal and hierarchical structures in the policy-making process, which can result in complex relationships between institutional actors and a lack of clarity in the policy formulation process (Cairney et al., 2019). For example, within the EU, there are multiple scales and layers of governance and authority, existing alongside NGOs and lobbyists (González, 2016). Thirdly, modelling institutional networks is confounded by the unstructured nature of the data that are available to policy actors (Lawrence et al., 2014). Data can be textual, and come in diverse formats, such as policy documents, grey literature, and research papers, which require institutional agents to understand natural languages including technical language. These challenges are not unique to this field; the simulation of human behaviour or ecological dynamics in the land system is similarly complicated, and solutions applied in these cases might be relevant here. Another similarity is in the value of such solutions, which cannot render a complicated system fully predictable but can reveal important dynamics stemming from behavioural processes (Davidson et al., 2024).

Large Language Models (LLMs), a form of artificial intelligence (AI), are based on numerous parameters that have been pre-trained on massive textual data and are designed to conduct natural language processes to understand and generate human-like text. The transformer architecture based on neural networks enables the LLMs to process sequences of text and contextual relationships between words (Vaswani et al., 2017). The text that LLMs produce is usually broken down into tokens, representing characters, sub-words or words (Minaee et al., 2024). LLMs have demonstrated strong language understanding and generation abilities and have emergent abilities such as multi-step reasoning that breaks down complex tasks into intermediate reasoning steps (Minaee et al., 2024). Hence, LLMs can be a powerful cognitive engine for autonomous agents that are able to sense the environment and act with regard to their own prescribed agenda (Wang et al., 2024a). LLM agents' ability to process and understand natural language allows them to synthesize information from various sources including unstructured data.

LLM agents provide high flexibility in modelling complex interactions between multiple decision-makers. Park et al. (2023) simulated an artificial village with 25 villagers powered by LLMs. The simulated villagers had heterogeneous personas and could interact with one another and their environment. These artificial villagers displayed believable, human-like behaviour and were able to organize a Valentine's Day party proposed by a user-controlled villager agent. Similarly, Qian et al. (2024) used LLM agents to simulate different roles in a software development team that is able to produce software cooperatively via a waterfall model. Further frameworks for dealing with many interacting agents have been emerging (see e.g., AutoGPT (Yang et al., 2023), AutoGen (Wu et al., 2023), AgentLite (Liu et al., 2024), MetaGPT (Hong et al., 2023)), which indicate the power of LLM agents in modelling complex relationships.

The aim of this study is to present a newly developed model, InsNet-CRAFTY, and explore the potential of modelling institutional networks in the land system using a state-of-the-art LLM agent approach. First, we identify the conceptual framework for implementing the institutional model and its coupling with a land use model. Specific tasks are assigned to the institutional agents to conduct numerical experiments. These experiments mainly serve proof-of-concept and exploratory purposes, which specifically include 1) investigating the LLM agents' logical coherence, 2) their contextual awareness within the network of institutions, and 3) their interplay with the programmed land use model. To aid in interpretation, the definitions and explanations of key

terminologies used throughout this paper are summarized in Table A1. We analyse the agents' textual output and numerical output to evaluate the believability of their decisions and the impact of their actions. We identify both opportunities and challenges for LLM agent applications in modelling institutional networks within the land system, which may provide useful insights into both model conceptualization and implementation for future research. This study also contributes to the broader field of institutional analysis in socio-ecological modelling, offering novel tools and methodologies for researchers and policy-makers.

## 2. Methodology

### 2.1 Model framework of InsNet-CRAFTY v1.0

We adopted the conceptual framework of a stylized, polycentric institutional network from González (2016), which offers a generic framework based on empirical evidence (e.g., peer-reviewed and grey literature) for Swedish forestry institutional actors. The key decision-makers included in the conceptual framework are the government, research suppliers, environmental NGOs, (forest) owner associations, and supranational institutions. The government has three levels, namely national, regional, and local authorities. González (2016)'s framework features both hierarchical and horizontal structures, offering rich components of a polycentric institutional structure while maintaining parsimony for computational modelling.

We further adapt González's (2016) framework through generalisation and abstraction to obtain the conceptual framework for this analysis (see Fig. 1). The framework maintains González's (2016) structural features, but the hierarchical governments are abstracted into two layers with one comprising a high-level institution and the other several independent operational institutions (representing different policy sectors), leading to greater governmental polycentrism. Additionally, two new actors are included - a law consultant and a narrative injector. In the context of modelling, we use agents to represent the modelled actors. A description of all of the LLM agents follows here.

*High-level institution*: The high-level institution can represent a supranational agency akin to the EU Commission. It sets the overall policy ambitions and constraints (e.g. budgets) that affect the decisions of the operational institutions. The high-level institution aims to achieve mid to long-term policy goals based on the information provided by the operational institutions, research suppliers, lobbyists, law consultant, and narrative injector.

*Operational institutions*: Operational institutional agents represent different policy sectors (e.g., agriculture, nature conservation, forestry, transport). They adopt and execute concrete policy instruments to influence the decisions of land user agents in order to achieve specific policy goals. Operational institutions can also submit action advocacies to the high-level institution to obtain budgets or permissions for certain policy actions.

*Lobbyists:* Lobbyist agents represent professionals who advocate for specific interests or causes (e.g. environmental NGOs and land use associations). Lobbyists in the model can observe the state of the land use system and form their own opinions about what should be changed to reach their own objectives. Their advocacy is considered by the high-level institution when making policy adjustments.

180

*Advisors:* Advisory agents can inform the high-level institution's policy-making using professional knowledge and skills. The framework considers two types of advisors: research suppliers and law consultants. The research suppliers observe land use changes and provide a description of the current and possible future land use states. They analyse and interpret both 185 numerical and textual data to support the high-level institution's decision-making. Law consultants offer information about existing laws, regulations, policies, etc., that legally underpin the high-level institution's policy actions; here we use EU policy documents to define these.

*Narrative Injector (optional)*: An agent whose absence does not affect the functioning of an 190 institutional network but can introduce highly unstructured exogenous disruptions into the model simulations through narratives (e.g., protest, war, unexpected disasters). The narratives can interact with all LLM agents in the model and can be injected at any point during the simulation. The narrative injector provides the means to explore the impact of shock and extreme events on the functioning of the institutional model.

195

Figure 1: Conceptual framework for InsNet-CRAFTY v1.0. The institutional network model is adapted from Gonzalez et al. (2016) and coupled with the CRAFTY land use model (Brown et

al., 2019). The hierarchical governments are abstracted into two layers with one comprising a high-level institution and the other several independent operational institutions to achieve greater governmental polycentrism.

Together with these institutional agents, we apply the CRAFTY land use model (Brown et al., 2019; Murray-Rust et al., 2014) to simulate land use changes in response to the institutional agents' interventions and potentially other drivers of change, e.g. socio-economic and climate change. The LLM agents form a stylized polycentric institutional model that can be implemented in a sequential order. For instance, CRAFTY can produce information indicating that both meat supply and protected areas (PAs) need to be improved to achieve better food security and nature conservation. This is an important example, as land is a finite resource, and meat production and PAs, among other things, compete for this land. Subsequently, the research supplier, operational institutions, and lobbyists collect and analyse the relevant data (e.g., the time series of meat supply and PA coverage) generated from CRAFTY. The data analysis serves as a basis for these agents to form different narratives that fit their distinct roles. The law consultant references policy and law documents to extract relevant information. The narrative injector may output a piece of news about an emergent incident. All these agents' output is eventually fed to the high-level institution, which considers the different stakeholders' positions and strives to make balanced decisions.

The high-level institution has concrete actions to influence the behaviour of the operational institutions, such as budget allocations and policy goal adjustments. It should be noted that the high-level institution cannot influence land users directly. Instead, the operational institutions have the autonomy to utilize their budgets to formulate and adjust policy instruments, such as subsidies, taxes, and administrative measures, to steer meat supply and PA coverage towards the target levels. In addition, the high-level institution does not have to be activated at the same frequency (in time) as the operational institutions, reflecting the asynchronous nature of institutional decision-making at different levels, particularly in the EU context. In Box 1, how this conceptual framework maps onto real-world policy institutions in the EU context is further explained with examples. Appendix A provides extra details and a technical description of the model's sequential processes.

Box 1. How this conceptual framework maps onto real-world policy institutions in the context of the European Union.

The conceptual framework presented here was inspired by the real-world mechanisms for policy delivery within the European Union (EU). Whilst the EU reflects a specific set of policy institutions and policy instruments, many of these concepts are transferable to other parts of the world with similar governance modes. In this box, we outline the relationships between the model components, especially the modelled agents, and their real-world counterparts that are outlined in Fig. 1.

The executive body of the EU is the European Commission, a supranational institution responsible for enacting new legislation. This is equivalent to the *high-level institution* in the model. The European Commission proposes new Directives and other policy measures that are ratified by the European Parliament, but which are then implemented by the member states (national governments). National scale implementation is done through various government departments that are usually responsible for a specific policy sector, e.g. agriculture, environment, research, etc. Within the model, these government departments are represented by the *operational institutions*, and the multiple instances of the *operational institutions* represent the different policy sectors. It should be noted that decision-making in the hierarchical governance structure is often asynchronous. The European Commission, as the supranational governing body, primarily focuses on long-term strategic policy goals (e.g., the European Green Deal, CAP reforms, etc.), setting overarching frameworks for Member States. In contrast, national and regional institutions make more frequent policy adjustments to ensure effective policy implementation and adaptation within their specific contexts.

Beyond the basic mechanism for policy implementation described above, policy institutions are influenced by a number of external bodies. Within the model, these are the *lobbyists* and the *advisors*. In the European context, lobbyists could include land owner associations with responsibility for the economic well-being of their membership. They also include environmental Non-Governmental Organisations (NGOs) that lobby for stronger environmental protection. For example, the European Livestock Voice group advocate for policies that support the sustainability and economic viability of the European meat industry, whilst organizations like EUROPARC seek to secure funding and favourable policies for the conservation and sustainable management of protected areas in Europe.

*Advisors* can include lawyers who support the legal aspects of policy development and implementation, as well as scientific researchers who provide policy institutions with knowledge to support policy development (at least in principle). It should be noted that the European Commission, a *high-level institution,* is also a very large research funder, providing financial support for policy-relevant research in universities and other research organisations across the EU.

## 2.2 LLM agent framework

To implement the institutional network model, the agents have to be equipped with a powerful cognitive architecture. Because of the extremely rapid evolution in the LLM field, a variety of ways to create LLM agents have been emerging (Sumers et al., 2024). Here we use the framework in Fig. 2 to represent the cognitive architecture of an LLM agent, which derives from the unified framework proposed by Wang et al. (2024a), and the LangChain framework (LangChain, 2024). This framework contains extensive elements and can represent a range of agent cognitive architectures from simple to complex, offering a unified structure to describe various agents in this research.

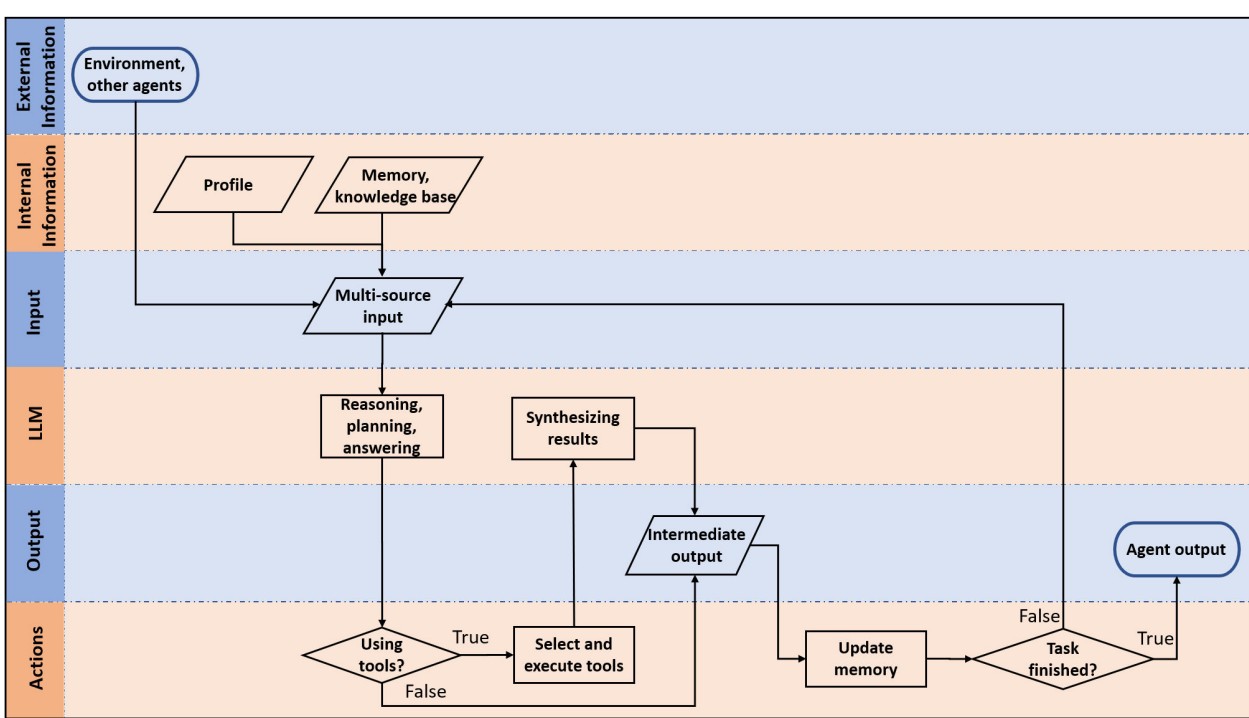

Figure 2: The cognitive architecture of a LLM agent. The core procedures of a LLM agent include the input, output, and the LLM. The agent's capability can be enhanced by integrating

sophisticated workflows such as memory, tool use, and reflection. Tools can be functions and
APIs coded in programming languages.

Although the complicatedness of different agents' cognitive architecture varies, the core of a LLM agent consists of a LLM and the LLM's input and output. The functionality of the LLM agent can
be enriched by incorporating more information into the input. Besides receiving external information from the modelled environment and other agents' responses, the LLM agent integrates internal information, such as a profile describing its identity, objectives, decision guidance, etc. An agent can also incorporate memory and a knowledge base into its input. Memory mechanisms indicate how LLM agents save and retrieve information. Memory is divided into short-term
memory and long-term memory. Short-term memory with high temporal relevance is embedded directly into the agent's prompt (the input directly received by an LLM). Knowledge and long-term memory relevant to a given decision-making context are extracted using Retrieval Augmented Generation (RAG) (Fan et al., 2024). Long-term memory reduces reliance on the context window (the maximum number of tokens an LLM can process at once) by enabling the
storage and retrieval of vast amounts of information beyond what the model can handle in a single input. For more technical details about LLM agent memory mechanisms, Zhang et al. (2024) provide a comprehensive survey on this topic. This multi-source information forms the input to prompt the LLM to generate reasoning and planning or to answer specific requests. If the agent is given a task to complete, the LLM helps to divide the task into small and achievable sub-tasks.

The capabilities of an LLM agent extend beyond text generation; it can actively execute sub-tasks and make decisions about the necessity of tools for task completion. In this context, "tools" refers to functions or APIs (application program interfaces) coded in programming languages, such as Python. For instance, a function might perform calculations that current LLMs struggle to handle
reliably on their own. An agent selects and employs appropriate tools to advance a task, as required. These tools process and organize results, which the LLM then synthesizes and outputs in natural language. Initially, these outputs are considered intermediate. The agent updates its memory by organizing and storing relevant inputs and outputs as necessary. It is worth noting that the intermediate outputs differ from the final outputs and work as step-by-step guidance that leads the
agent to reach a refined final answer. These step-by-step intermediate outputs can be generated by the agent spontaneously or given by the end-user through the prompt (see Table B1 for example). Subsequently, the agent evaluates whether the tasks are complete to decide whether to produce the final output or to continue processing with updated memory.

The decisions on which tool should be used and when to output a final answer are highly autonomous. However, the end-user can also influence this process by explicitly instructing the agent on tool selection and the tasks it needs to execute before ending the "thinking" processes. Tool selections and the ending timing are informed by the agents' textual output that contains several "cues", such as the names of the tools and a special string that indicates the end of the
agent's thinking process. In addition, many agent frameworks, such as LangChain, set operational constraints on how many loops or how much time the agents can use before giving final outputs. This serves as a safety measure in case the agent is trapped in meaningless loops. The LangGraph (a framework within the LangChain ecosystem) documentation gives more technical details on agent workflows (LangGraph, 2025).

## 2.3 Experimental Settings

Since examining logical coherence and contextual awareness of the LLM agents requires meticulous manual assessment of the agents' textual output, a crucial guideline for setting the experiments is to be simple and straightforward. The settings related to the LLM agent behaviours are structured in Table 1, including their requirements for inputs, actions, outputs, goals, policy instruments, and LLM versions. The prompt templates of the agents are provided in the tables in Appendix B. The settings pertinent to model workflow or parametrisation are detailed as follows.

*The CRAFTY land use model:* CRAFTY models land use dynamics by having land user agents - referred to as Agent Functional Types (AFTs) - compete for land (Murray-Rust et al., 2014). Each AFT employs the capitals available on its land to produce a mix of ecosystem services. The marginal utility of an ecosystem service is related to the gap between its demand and actual production, which motivates AFTs to make decisions regarding land turnover. This utility-based competition can give rise to an emergent trend at the system level where the modelled ecosystem service supply approaches the demand level (Zeng et al., 2025b). We set up the land use model according to CRAFTY-EU (Brown et al., 2019) and parametrized it with the data for the RCP2.6-SSP1 climatic and socio-economic scenario (Brown et al., 2019). Under this scenario, the land use model produces a gradual and steady increase in ecosystem service supply driven by the changes in demand (Zeng et al., 2025b), which provides a relatively simple baseline and can mitigate complications in analysing the LLM agents' impact (Zeng et al., 2025c). Seventeen AFTs generate a variety of ecosystem services, including meat, crops, timber, and others (Brown et al., 2019). Policy interventions are represented by mechanisms that either enhance or diminish an AFT's utility (see Eqs. (C9) and (C10)), thereby guiding the overall system toward policy targets. The CRAFTY-EU model uses a map of European countries at a 5-arcminute resolution (Brown et al., 2019). The scenario simulation covers the period from year 2016 to 2076 based on the available data (accessible on Zenodo (Zeng, 2024a)). Each year is simulated as one iteration.

*Operational institutions*: We narrowed the scope of the modelled actors by specifying their roles and responsibilities. Instead of integrating a diverse array of operational institutions with a wide range of policy objectives and tools, we incorporated two distinct operational institutions focused on different policy sectors: an environmental institution and an agricultural institution. The former prioritises environmental protection with a specific aim of expanding protected areas for nature conservation, while the latter focuses on meat production to ensure food security using economic policy instruments, such as subsidies. Since meat consumption is a major driver of deforestation, greenhouse gas emissions, climate change, and biodiversity loss (Djekic, 2015; Machovina et al., 2015, Petrovic et al., 2015), this experimental setting creates a conflicting context for the two operational institutions competing for limited budgets and potentially land space, to fulfil their respective policy objectives. This stylized setting reflects a real-world conflict in EU land use, where livestock farming and nature conservation compete for finite land resources, and policy interventions are a major contribution to resolving this competition (Acs et al., 2010; Maes et al., 2012).

It is worth noting that the operational institutions are modelled as "hybrid agents". In addition to the LLM-driven functionalities presented in Table 1, they utilise rule-based decision-making powered by a Fuzzy-PID controller mechanism (Zeng et al., 2025b). They can use expert knowledge encoded in IF-THEN rules to adjust policies in response to the discrepancies between

policy targets and the actual land use outcomes, such as meat supply and PA coverage. This hybrid design resonates with the non-routine and routine activities in organisational behaviour studies (Simon, 2013). The non-routine activities refer to the operational institutions' influence on the high-level institution by using natural language to propose policy goals and budget adjustments; while routine activities include the operational institutions' policy interventions executed within the programmed land use model. Figure A1 shows how these hybrid agents are integrated into the modelling processes. The operational institutions' rule-based activities include policy evaluation and adaptation (Eqs. (C1) – (C5)), budget use (Eqs. (C6) – (C8)), implementation (Eq. (C10)), and their interactions with the high-level institution (Eqs. (C11) – (C15)). The related numerical settings are given in Tables (C1) – (C2). The code is available on Zenodo (Zeng, 2024b).

*Policy-making frequencies*: As previously stated, the high-level institution and the operational institution operate on different time cycles. Here, the high-level institution is activated every ten iterations, while the operational institutions adjust their policies every two iterations via rule-based decision-making, representing routine activities and a more frequent response in policy adjustments. This frequent adjustment reflects the agility of the operational institutions compared to the slower, more deliberative pace of the high-level institution. As ten is a multiple of two, the operational institutions' LLM components are enabled whenever the high-level institution is activated, allowing them to communicate in natural language.

*Initial targets and budget allocation*: We set the initial target meat supply as 1.2 times the initial meat production level, and the target of PA coverage as 10% of the total land area. These parameters give the operational institution agents slightly higher but achievable initial targets to pursue. The initial budget allocation between the two operational institutions is set at 50% each. We chose an equal initial budget allocation because 1) the LLM agents have no prior knowledge about the simulation environment within the first ten iterations; 2) thus an unequal initial budget allocation would be harder to justify at any specific level than an equal one; 3) an equal initial budget allocation offers a simple baseline in contrast to the agents' ensuing budget adjustments from the tenth iteration, which is an important LLM agent behaviour the experiments are intended to investigate.

*LLM versions*: Accessing LLMs through APIs is convenient, but it has limitations and requires strategic use. LLM API providers typically charge based on the length of inputs and outputs measured in tokens, which are basic units encoded to train LLMs. Rate limits are another factor imposed by APIs that constrains the frequency of LLM responses. One method to mitigate the rate limits is to force the model to "sleep" periodically, which slows down the simulation speed. To improve the token cost and mitigate rate limits, we used a combination of gpt-4o (OpenAI, 2024) and Llama-3-70b-8192 (Llama, 2024) to drive the LLM agents. At the time when these experiments were conducted, Llama-3-70b-8192 was accessed through the Groq platform (Groq, 2024), which offered free services. The agents called the LLMs alternatively to avoid sending requests to a single LLM too frequently.

*Structured output*: Because the LLM agents are coupled with a programmed land use model, the LLM agents need the capability to structure data in a designated format, otherwise, the programmed land use model would not parse the data, which could disrupt the simulations. Here, we prompted the high-level institution to structure the adjusted policy goals and budget allocation

in a JSON format, on which many current LLMs have been fine-tuned, to format the output of the high-level institution. Despite this, there is no guarantee that LLMs always follow the formatting requirements strictly. We employed three layers of mechanisms to increase the probability of deriving correctly formatted data, including re-prompting the LLM and using regular expressions (Li et al., 2008) to identify JSON structure, and manually formatting if necessary.

Table 1 Experimental settings of the LLM agents. The LLM agents with actions were built using the LangChain library (LangChain, 2024).

| Agent Type | Agent Role | Input | Action | Output | Remarks |
|---|---|---|---|---|---|
| Advisors | Research supplier | 1) CSV file containing data from CRAFTY 2) Profile (Table B1) | Writing and executing Python code to analyse the data. | 1) Unstructured text to inform other agents 2) Intermediate output - stepwise output that leads to a final response. | 1) Goal: Analysing and interpreting the data generated by CRAFTY. 2) LLM version: gpt-4o |
| | Law consultant | 1)Document containing EU laws, policies, regulations etc. 2) Profile (Table B6) | Using RAG to retrieve relevant information from a selected set of EU policies. The data are available on Zenodo (Zeng, 2024a). | Unstructured text to inform the high-level institution's decision-making | 1) Goal: Extracting relevant information from a knowledge base to inform the high-level institution's legal actions. 2) LLM version: Llama-3-70b-8192 |
| Lobbyists | Environmental NGO | 1) Research supplier's output 2) Profile (Table B2) | None | Unstructured text to lobby the high-level institution | 1) Goal: Lobbying the high-level institution to prioritise nature conservation. 2) LLM version: Llama-3-70b-8192 |
| | Land user associations | 1) Research supplier's output 2) Profile (Table B3) | None | Unstructured text to lobby the high-level institution | 1) Goal: Lobbying the high-level institution to prioritise meat industry development. 2) LLM version: Llama-3-70b-8192 |
| Operational Institution | Environmental institution | 1) CSV file containing data from CRAFTY 2) Profile (Table B5) | Writing and executing Python code to analyse the data. | 1) Unstructured text to inform the high-level institution 2) Data for CRAFTY code | 1) Goal: Striving to acquire budget to support PA expansions to reach the target PA coverages. 2) Policy instrument: PA designation. 3) LLM version: Llama-3-70b-8192 |
| | Agricultural institution | 1) CSV file containing data from CRAFTY 2) Profile (Table B4) | Writing and executing Python code to analyse the data. | 1) Unstructured text to inform the high-level institution 2) Data for CRAFTY code | 1) Goal: Striving to acquire budget to support meat production to reach the target meat supply level. 2) Policy instrument: economic measures (e.g., taxes and subsidies) 3) LLM version: Llama-3-70b-8192 |
| High-level (e.g., supranational) Institution | High-level institution | 1)Unstructured text from all other agents 2) Profile (Table B7) | None | 1) Unstructured text 2) Policy goal and budget allocation adjustments in JSON structure for CRAFTY to parse | 1) Goal: Making policy adjustments based on multiple stakeholders. 2) Policy Instrument: Administrative orders to adjust the operational institutions' policy goals; financial measures to allocate budget between the operational institutions. 3) LLM version: gpt-4o |

| Others | Narrative injector | None | None | None | The narrative injector agent is not included in the results reported here for simplicity and in order to maintain the system's full autonomy. |
|---|---|---|---|---|---|

## 2.4 Analytical Methods

### 2.4.1 Manual examination

The LLM agents' logical coherence and contextual awareness were the primary focus of the analysis. It is necessary to scrutinize their textual output cautiously. One of the reasons for doing so is that all LLMs hallucinate to some extent (Banerjee et al., 2024), meaning they can produce factually, logically, or contextually incorrect responses in a confident tone (Ji et al., 2023). There is also no standard answer to policymaking problems. Automated rule-based evaluation of their textual outputs can offer very limited utility. Therefore, manually evaluating responses is unavoidable. Specifically, we examined logical coherence by looking into the LLM agents' policy decisions and arguments to see whether there was a contradiction or flawed reasoning; contextual awareness was assessed according to whether their responses aligned with their roles, updated information, and continuity of discourse. In addition, as agents are composed of LLMs coupled with peripheral workflows (see Fig. 2), it was also necessary to identify any malfunctions caused by the workflows.

In a computer program, an "error" typically refers to a deviation from the intended functionality or an outcome that violates the system's logical rules or design specifications. We broadly defined errors related to the LLM agents in terms of logical coherence, contextual awareness, or workflows as erroneous behaviours. Not all erroneous behaviours are negative because some of them might capture important aspects of human behaviours. We can define their activities regarding reasoning and mistake-making that resemble real-world humans or human organizations as believable behaviours. Occasionally, LLM agent behaviours seem bizarre but do not necessarily indicate errors, which can be seen as counter-intuitive behaviours.

Erroneous behaviour of the LLM agents could affect the robustness of the model and the approach used. Robustness here refers to the ability of a system to function effectively despite errors. This type of robustness does not imply statistical robustness (which requires sensitivity analysis) but rather operational resilience. As robustness here reflects how well the model performs with the existence of agent erroneous behaviours, it is worth careful investigation. However, since the forms of erroneous behaviours are not definite or foreseeable, manual examination is necessary.

### 2.4.2 Word frequency and word graph analysis

Word frequency and word graph analysis are important techniques for examining text patterns. These tools can help identify key themes, relationships between terms, and underlying structures in a dataset, which are well suited for analysing the lobbyist agents who have high autonomy to defend their interests. Word frequency analysis counts how often specific words appear in a text. We visualized the word frequencies and used Zipf's law to fit word frequency distributions (Piantadosi, 2014). Zipf's law can be expressed as $f(r) = k/r^s$, where f(r) is the frequency of a

word; r represents the rank of the word according to its frequency; k and s are parameters. A larger s indicates a set of words distributed more unevenly.

Word graph analysis was used as a further step, to explore how words connected to each other. This method builds a network where words act as nodes and their connections form edges. The edges are calculated based on the words' co-occurrences in a "sliding window" with a prescribed length. The sliding window scans the given text from the beginning to the end and records the co-occurrences of words. The edges can have thickness proportional to the frequencies of word co-occurrences. As a result, word graphs help uncover semantic structures and reasoning patterns, showing how ideas are interconnected.

### 2.4.3 Numerical output analysis

The numerical output analysis was focused on the time series of budget allocations and policy goal adjustments conducted by the high-level institution, as well as meat production, PA coverage, and budget surpluses of the two operational institutions. It is noteworthy that budget surpluses were estimated as the budget needed to implement a policy minus the budget an operational institution actually possessed (see Eqs. (C11) – (C15) for details). Budget surpluses can be either positive or negative, respectively indicating over- and under-supply of financial support by the high-level institution.

## 3. Results

### 3.1 Policy actions and outcomes

The results shown in Fig. 3 (a) illustrate that the high-level institution increased the policy goals for PA coverage gradually across the simulated time period, which resulted in a stepped pattern of PA growth. This reflects the periodic activation of the high-level institution as described previously. The actual PA coverage shows a tendency to approach the target PA coverage. The eventual policy goal was set at approximately 30%, which drove the actual PA coverage to approach this level. In some years (between 2046 and 2076), the actual PA coverage reached the target and then remained almost unchanged for several years until the high-level institution raised the targets again. At each stage, the high-level institution's policy goal adjustments were aligned with the environmental institution's capability to influence PA coverage, as apparent in the rapid closure of gaps between target and actual PA coverage across the time period. However, the budget surplus remained positive and grew over time, which indicated over-funding by the high-level institution.

Similar to the policy goal adjustments in the PA coverage, Fig. 3(b) shows that the high-level institution increased the target level of meat production periodically and gradually, resulting in a stepped growth over time. Meat supply was positively correlated with the policy goal, and although meat supply was not able to reach the policy goal, the goal-supply gap was limited within a relatively small range. In 2065 meat supply plateaued, while the ensuing policy goal adjustment at 2075 was still increasing. In contrast to the environmental institution, the agricultural institution underwent increasingly severe budget restrictions.

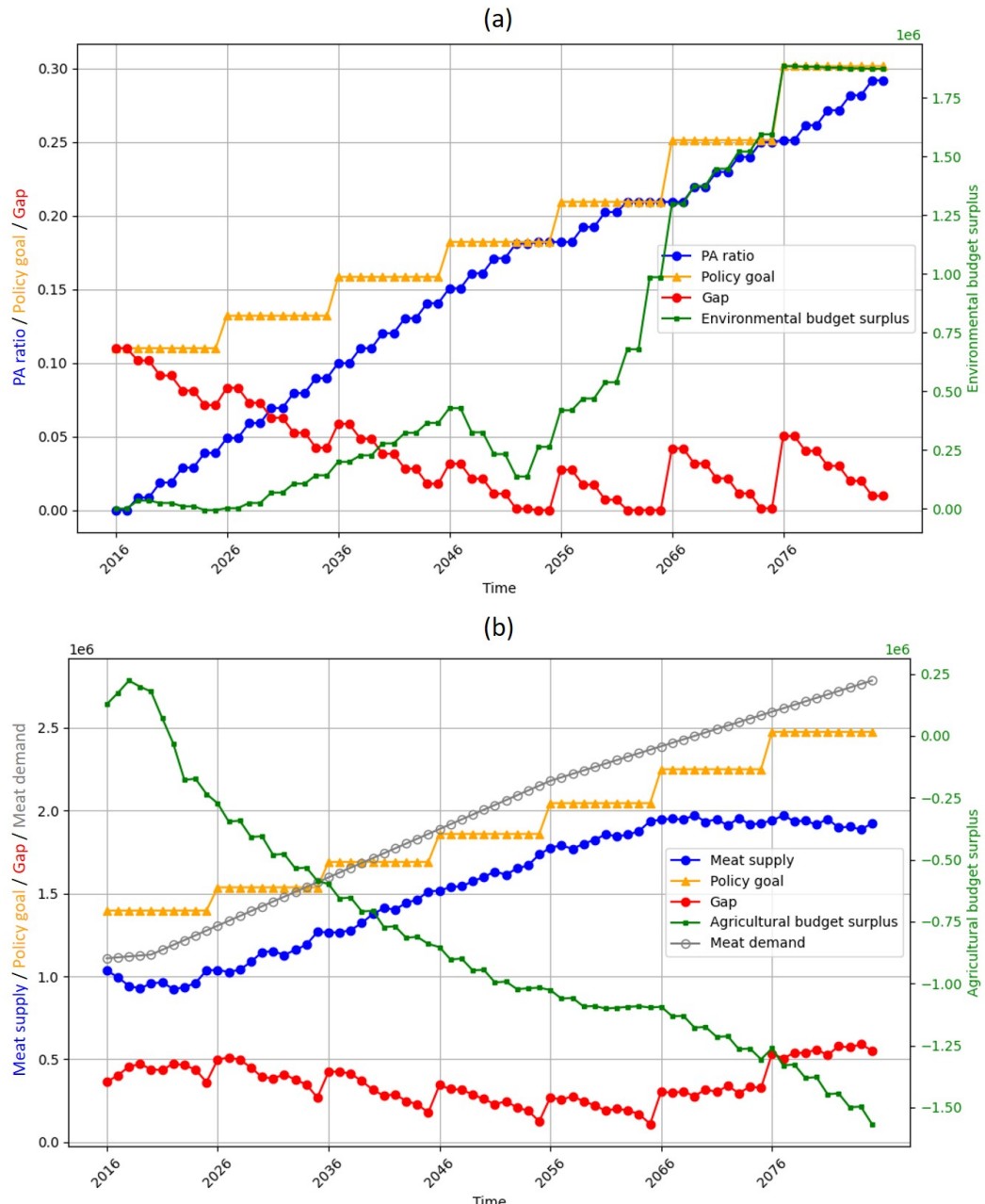

Figure 3: Policy goal adjustments, budget allocation, and their impacts for a) the environmental operational institution agent and b) the agricultural operational institution agent. "Gap" indicates the difference between the policy goal and corresponding land use outcomes (e.g., PA coverage and meat supply)

Figure 4 shows the budget allocated by the high-level institution. In the first ten years (from 2016 to 2025), the budget allocation between the two operational institutions is 50/50 by default. However, it can be seen that the high-level institution shows a tendency to avoid radical changes in budget allocation. Despite the agricultural institution's lack of budget, the budget allocated between these two operational institutions was 60/40 from 2026 to 2045, 30/40 from 2046 to 2055, and 45/55 from 2076 to the end of the simulation. The high-level institution chose to allocate more

budget to the agricultural institution in only twenty iterations, even though the latter's budget deficit occurred very early (before 2026). The twenty iterations include a counter-intuitive budget drop for the environmental institution in 2046, resulting in a total budget of 0.7 instead of 1.0, as in the other iterations.

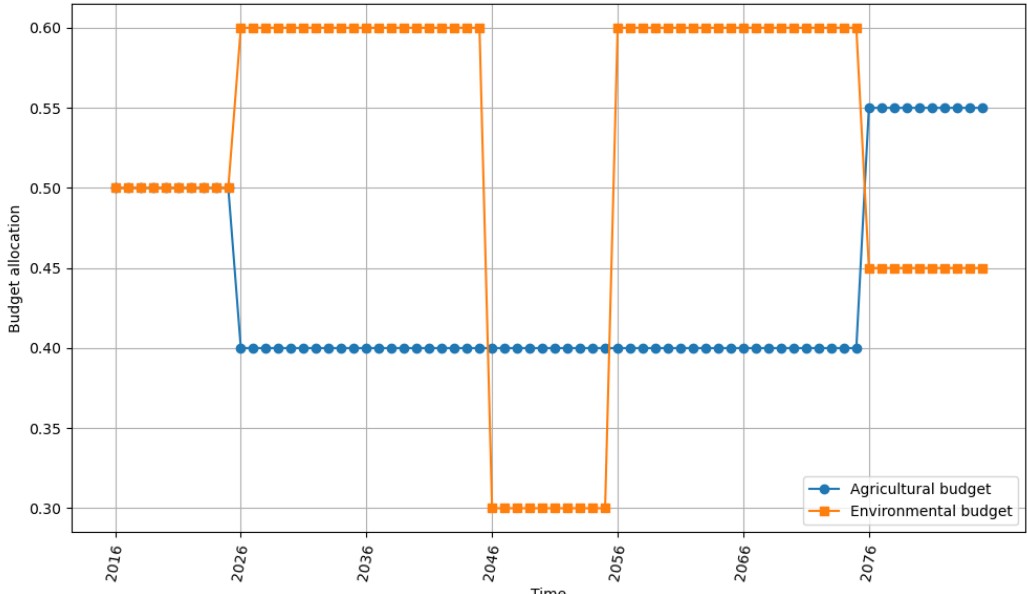

Figure 4: Budget allocation between the agricultural and environmental operational institution agents.

## 3.2 Textual output

The LLM agents' output contained 19808 words (28778 tokens) and 48 plots. We summarised the textual output that demonstrates the behavioural patterns of the agents, while also highlighting counter-intuitive or potentially erroneous agent behaviours. An interactive visualization of the LLM agents' textual output over time is available at Zeng and Byari (2025).

### 3.2.1 The advisors' output

The textual output demonstrates that the research supplier followed the instructions in its prompt (Table B1) to complete various tasks in the correct order, including the intermediate steps of checking missing values in the data, interpreting the trend of meat supply and demand, and analysing the discrepancy between policy goals and actual outcomes. Figure 5 briefly illustrates the workflow of the research supplier agent. It made plans, executed the plans step by step and interacted with tools. The tools offered returned values to form intermediate output, which was fed back to the LLM. A final output was produced based on the intermediate outputs. In some of the final outputs, the agent attached a note at the end of the output as a reminder of the applicable scope of the analysis, e.g., "Note: The above insights are based on the analysis of the provided data and may not be generalizable to other contexts". In 2066, the agent encountered an error – "Agent stopped due to iteration limit or time limit" – indicating the actions the agent needed to take exceeded the set maximum execution time.

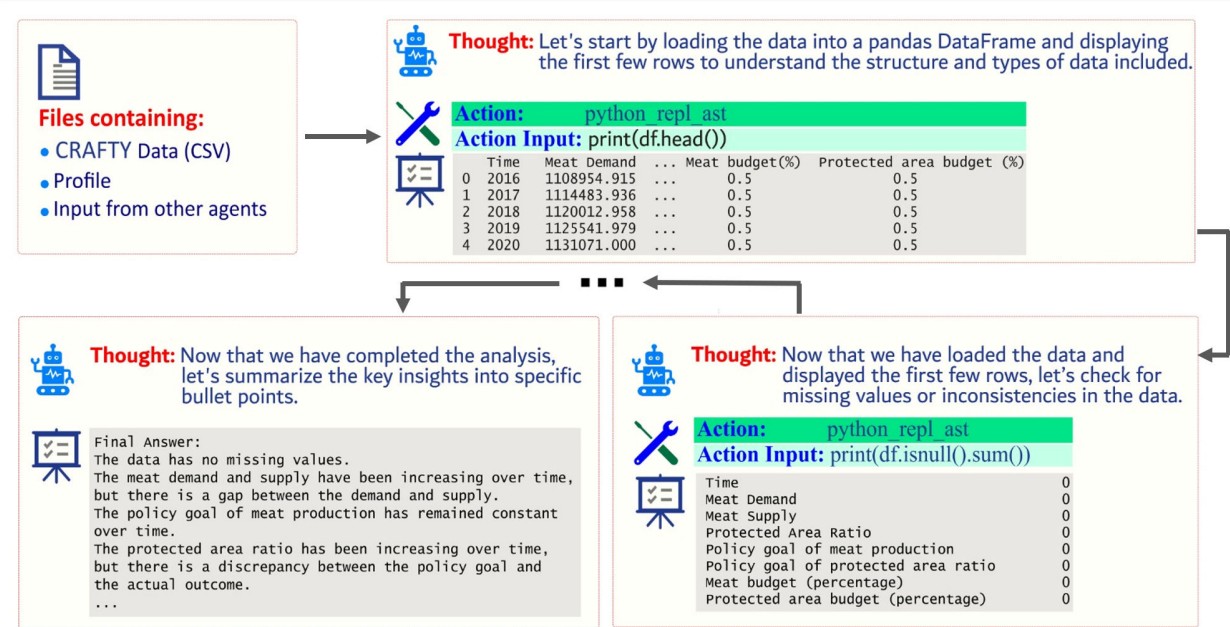

Figure 5: The workflow of the research supplier agent. The agent took the initial input to generate a thought to decide what actions should be taken to analyse the data. Then, it executed the action by calling a function, which in turn produced the intermediate results. These results served as a part of the updated input to let the agent generate a new thought for the next iteration. After several iterations of thought-action-output loops, the research supplier agent produced a final interpretation of the data.

The law consultant emphasized six critical aspects to influence the decision-making of the high-level institution, based on the available knowledge base. These aspects include "biodiversity and ecosystem restoration targets," "agricultural production and environmental impact," and "climate change mitigation". The agent not only highlighted these issues but also cited relevant laws, policies, and directives. Furthermore, the agent elaborated on the implications of these legal and policy frameworks for the high-level institution's policy-making processes. For example, in discussing "biodiversity and ecosystem restoration targets," the law consultant noted that "the EU Restoration Law mandates the restoration of at least 20% of the Union's terrestrial and marine areas by 2030, and all ecosystems in need of restoration by 2050." This law was interpreted to mean that "a significant portion of the budget should be dedicated to protected areas to meet these objectives." It was observed that the law consultant agent produced the same output repeatedly over several iterations, reflecting stagnation due to the absence of new inputs that could prompt different responses.

### 3.2.2 The lobbyists' output

The environmental NGO generated a variety of arguments for prioritising protected area establishment over meat production. For instance, in some years the agent highlighted the urgent need for nature conservation, the impact of meat production, or the necessity of budget increase. In 2066, the environmental NGO agent did not receive information from the research supplier due to the error mentioned above. However, this error did not paralyze the simulation. Instead, the LLM agent stated "I apologize, but it seems like there is no information provided. However, as a

representative of an environmental NGO, I can still provide some general bullet points to lobby a high-level public policy institution to prioritise nature conservation." Without basing its arguments on data, the agent emphasized the economic benefits of nature conservation, the importance of PAs to climate change mitigation and adaptation, human health and well-being.

The land user association agent also utilised background information and the data interpretation provided by the research supplier agent to lobby the high-level institution to prioritise meat industry development. For instance, this agent highlighted economic growth, job creation, food security, and alignment with policy goals. When the output from the research supplier agent was missing, it gave more general bullet points to lobby the high-level public policy institution, including emphasizing the meat industry's economic benefits, food security, rural development, innovation and technology without using any data from CRAFTY.

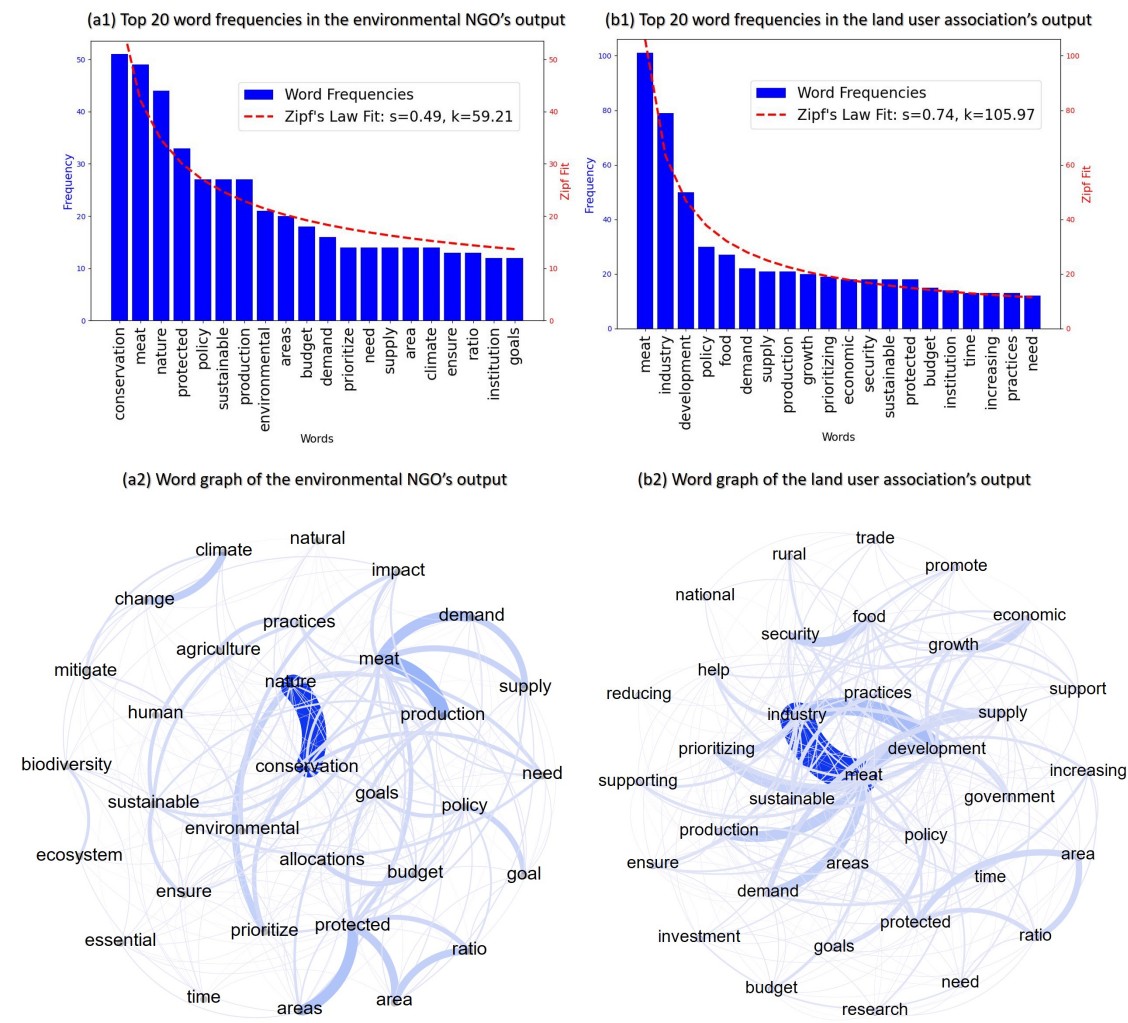

Figure 6: Word frequencies and word graphs derived from the lobbyists' output. The dashed red lines in (a1) and (b1) are derived by fitting Zipf's law distribution to the word frequency distributions. It can be seen in (a1) $s = 0.49$ for the environmental NGO's output and in (b1) $s = 0.74$ for the land user association's output, reflecting the two agents' different approaches

in formulating their arguments. The word graphs only display nodes that have more than thirty links, in order to maintain visual clarity.

To better visualize how the lobbyists formulate their arguments, Fig. 6 illustrates the word frequencies and relationships through word graphs derived from their outputs. The analysis reveals a less prominent skew in the frequency distribution of the top 20 words used by the environmental NGO compared to those of the land user association. The environmental NGO frequently emphasized the term "conservation" and notably the word "meat." Its discourse primarily focused on two aspects: the environmental threats posed by meat production and the critical importance of conservation efforts. This concern was underscored by the research supplier's data interpretation showing a widening gap between meat demand and supply. In contrast, the land user association highlighted the development of the meat industry and food security, without opposing the expansion of protected areas. Instead, the land user association consistently advocated for sustainable meat production practices, which they argued would support their request for an increased budget.

### 3.2.3 The operational institutions' output

The agricultural institution's outputs consistently addressed the discrepancies between the meat production policy goals and the actual outputs, alongside recurring budget challenges. This agent repeatedly emphasized the necessity of addressing budget deficits, advocating for more efficient budget allocations and increased financial support to meet production goals. Key recommendations included increasing budget allocations to bridge the gap between policy goals and actual outcomes, setting realistic policy goals that align with current capacities, and enhancing sector productivity through various initiatives, e.g., farmer incentives and sustainable practices. Additionally, the institution suggested establishing a robust monitoring and evaluation framework to regularly assess the effectiveness of policies and adjust as necessary. A holistic approach was advocated to balance increased production goals with budget constraints, thereby boosting food security, improving farmer livelihoods, and ensuring financial well-being.

The environmental institution consistently highlighted a gap between the current state of protected areas and policy goals over the decades, emphasizing the need for increased financial support and a higher priority for protected area establishment to achieve biodiversity conservation and pollution reduction. Recommendations include raising the PA goals incrementally each year, improving governance, enhancing community engagement, and specifically allocating a substantial percentage of budget surpluses to facilitate the expansion of PAs. These steps were deemed crucial by this agent for reaching net-zero targets and effectively managing biodiversity conservation amidst evolving environmental challenges. However, the agent mistakenly used mean values to describe the time series, which generated misleading outcomes. For instance, in the year 2076, the actual protected area is 25.14% and the target is 30.17%; however, the environmental institution used the mean values of 13.44% and 17.40% respectively to inform the high-level institution about the current situation. This error did not, however, qualitatively change the need to expand protected areas.

### 3.2.4 The high-level institution's output

The high-level institution employed a systematic and analytical approach to decision-making, consistently integrating stakeholder feedback across several sectors to refine policy goals and

allocate budgets effectively. This process involved a detailed analysis of input from agricultural and environmental institutions, NGOs, and industry associations. Key actions include adjusting policy goals and redistributing budget percentages to better support the targeted outcomes in meat production and environmental protection. The institution regularly adjusted its strategies, intending to bridge the gaps between current outcomes and policy objectives, focusing on sustainability, economic stability, and nature conservation. However, the output of the high-level institution was sometimes inaccurate. For instance, the high-level institution's analysis only included information from all six of the LLM agents in 2036 and 2056 with the law consultant and/or the research supplier's inputs occasionally being missed.

## 4. Discussion

### 4.1 Believable behaviour of the LLM agents

Building upon the pre-trained LLMs, the institutional agents modelled in InsNet-CRAFTY exhibited diverse human-like reasoning and actions. The agents' behaviour was guided using prompts in natural language, which gave the model developers high flexibility in creating the agents' autonomous behaviours. This flexibility facilitated the modelling of the complex relational structures among the agents. Given appropriate profiles, the agents were clear about their identities and relationships with others, demonstrating consistent role-specific decision-making. The LLM agents also showed an ability to handle the qualitative and unstructured information generated by the lobbyists, law consultants, and advocacies from the operational institutions. The capability of function calling (e.g. writing and executing computer code) further improved the agents' autonomy, enabling the latter to deal with more complex tasks, such as data analysis and knowledge retrieval. These capabilities suggest that LLM agents have a unique potential to overcome the key challenges in modelling institutional networks.

Besides these apparent strengths, the LLM agents showed flawed but understandable behaviour when faced with key real-world challenges, such as conflicting objectives, uncertainty and imperfect (or absent) information. The budget allocation was a major output of the high-level institution, which reflected competing claims for a limited resource. An impromptu suggestion by one of the lobbyist agents indicated that money be transferred to research to better understand policy impacts. The textual output shows that this suggestion was responsible for the counter-intuitive sudden drop in the total budget when the high-level institution allocated 30% of the total budget to "other initiatives and programs". These dynamics could allow many important policy processes to be investigated, including observed differences between budgeting systems based on plurality, proportional representation or public participation, in which information inputs and decision-making powers vary substantially (Feindt, 2010; Hallerberg and Von Hagen, 1997; Lee et al., 2022).

The numerical results regarding the gap between target and actual protected area coverage demonstrate how environmental institutions respond to policy objectives. Initially large, this gap gradually diminished until nearly closing. When the gap was minimal, actual coverage remained stable, suggesting limited institutional intervention during this period. When higher-level policy changes subsequently created new gaps, the environmental operational institution responded by expanding protected areas, creating a stepwise pattern of adjustment, illustrating the responsive relationship between policy targets and the operational institutions' policy adaptations. However,

the environmental institution in the simulations was over-funded, while the agricultural institutions were struggling with an inadequate budget. We found that this imbalance was prompted by the environmental institution and the research supplier misleadingly informing the high-level institution that PA coverage was positively correlated with budget surplus.

Indeed, both the two operational institutions insisted that their respective policy targets (PA coverage and meat production) should be increased because those targets were positively correlated with other desired outcomes. Nevertheless, advocacy efforts were not equally effective. The environmental NGO's arguments were backed up by urgent environmental concerns, and outweighed the more economically focused arguments of the land user association. This imbalance might derive from the LLMs' training data being influenced by present-day social norms and highlights the potential for biases to be embedded within the agents' roles. LLM biases have been well documented in the literature (see e.g., Zhou et al., (2024)) and can be rectified by prompt design and fine-tuning (Taubenfeld et al., 2024; Tao et al., 2024). These also of course reflect policy biases in reality, however, where norms, power relations, communication and urgency all affect policy priorities, and potentially allow exploration of approaches to mitigate these issues in differing policy contexts (Barnett and Finnemore, 1999; Sinden, 2004; Yami et al., 2019).

The resultant budget allocation diverges from intuitive expectations observed in real-world cases, where environmental funding is typically insufficient (Cosma et al., 2023; Waldron et al., 2013). This divergence is partly due to the assumed parameterization (see Table C1): the budget settings are designed to ensure that the budgets required for different policies are comparable in scale, rather than being precisely calibrated to force the high-level institution into making trade-offs between fostering meat production and expanding protected areas. Consequently, the primary challenge for the high-level institution is not to choose one policy target over the other but to coordinate policy goals with sufficient budgets to enhance overall efficiency. The mismatch between modelled and real-world budget allocation may also actually indicate unrealistically rational behaviour of the modelled agents, in recognising the urgency of conservation to an extent that governments rarely do.

In any case, the results capture the challenges of redistributing resources in multi-actor governance contexts, such as in the EU's Common Agricultural Policy (CAP), where competing priorities frequently lead to compromise-based rather than optimal budget allocations (Daugbjerg & Feindt, 2017). A more successful strategy in the simulations might have been a radical reallocation of funds, such as shifting the majority of the budget toward the underfunded agricultural sector at an earlier stage. However, the failure to pursue this strategy does accord with reality, where bounded rationality (Simon, 1972; Gigerenzer and Goldstein, 1996; Jones, 2003) and policy-making by 'muddling through' (Lindblom, 1989) are often apparent in complex systems.

## 4.2 Challenges of implementing LLM agents

Along with the advantages of the LLM agent approach in simulating institutional networks, erroneous behaviour was also apparent. Typical causes of errors were flaws in agent workflows and LLM hallucinations (Ji et al., 2023; Perković et al., 2024; Yao et al., 2023). The research supplier agent's occasional failure to output data interpretation was caused by a flaw in the agent workflow that generated the error "Agent stopped due to iteration limit or time limit." This error could easily be avoided by increasing the permitted number of iterations that an agent needs to

execute a complex task, although it had the advantage of forcing the other agents to proceed with imperfect (out-of-date) information, as is often the case in real-world contexts (Arnott et al., 1999; Callander, 2011; Neri and Ropele, 2012).

Unlike the research supplier, the operational institutions were not given specific data analysis instructions. This led to an unexpected outcome – the operational institutions tended to use mean values to describe the latest state of the time series and so provided misleading information to the high-level institution. Such erroneous behaviour can be categorized as hallucination because the agent used plausible-sounding words to express nonsensical information (Ji et al., 2023). This erroneous behaviour could be mitigated by using more specified instructions in the prompts to guide agents' reasoning or designing extra mechanisms to detect and rectify the LLM's response (Tonmoy et al., 2024). However, addressing LLM hallucination is challenging, and there is no standard solution so far.

For large-scale land use models, another crucial challenge is an LLM's limited context window. Here, the high-level institution had to consider all the other agents' output to make decisions. The resultant input could be very lengthy. Issues might arise if the input exceeds the maximum number of tokens (namely the size of the context window) that an LLM could digest. Technically, the size of an LLM's context window is generous. However, research shows that the reasoning performance of LLMs drops notably as the length of prompts increases even if the length is far less than the technical maximum (Levy et al., 2024). In the real world, institutional networks are far more complex than those in this model, and it is not unusual for high-level institutions to be overwhelmed by the information that they need to assimilate, and to use information selectively as a result (Bainbridge et al., 2022; Fischer et al., 2008; Rich, 1975). The limited size of the context window can therefore be seen as a feature that reflects human decision-makers' bounded rationality and information processing capabilities or preferences, as well as the imperfect nature of much information used for decision-making (Neri and Ropele, 2012). However, whether it is preferable to model such a feature in a controlled manner or to rely on the result of a technical limitation is debatable. The technical limitation could be mitigated by using summarized input or including memory mechanisms with retrieval methods (Modarressi et al., 2023; Zhong et al., 2024; Zhou et al., 2023), although these approaches all require extra effort in designing peripheral agent workflows (Li et al., 2024; Wang et al., 2024b).

In contrast to the above limitations, the data formatting issue could be more cumbersome to handle. Given the stochastic nature of LLMs, there is no guarantee that LLMs can always accurately format their output. This leaves an extra task to design peripheral workflows to secure the format. However, as the model scales up to integrate numerous LLM agents, the likelihood of glitches arising from their interaction with existing programmed models also increases. In such cases, a robust and well-designed error-handling mechanism becomes essential. Error-handling mechanisms in terms of formatting can vary widely depending on specific modelling purposes. In large-scale simulations driven by LLM agents, where the focus is on system-level emergent patterns rather than individual agent actions, a straightforward guideline is to, at least, prevent malfunctioning agents from crashing the entire simulation. Errors should be quietly logged in the backend for the simulation to continue uninterrupted. Additionally, the error rate should be tracked and reported to help modellers examine if the overall result remains valid. For example, in a system with 1,000 LLM agents generating outputs every iteration, a 5% error rate might be tolerable if it

has no significant impact on the overall patterns being studied. However, as there is no one-size-fits-all solution so far, explicitly presenting how errors are handled and the rate of errors in simulations should be helpful for research transparency.

## 4.3 Paradoxical robustness

The results implied a paradoxical relationship between the LLM-based institutional network model's error proneness and error tolerance, which could enhance the understanding of the robustness of multi-agent systems. For instance, with multiple institutional actors joining the system, the chances of erroneous behaviour increase since every single decision-maker has some probability of producing errors. These errors could also be transmitted within the network and 800 affect other agents' decision-making, which, to some extent, corresponds to real-world policy-making. However, with the interaction of multiple agents, no single agent nor their erroneous behaviour had sufficient influence to determine the behaviour of the whole system. The missing output from the research supplier did not lead the system to generate a cascade of unusual behaviour nor did it crash the simulation. The high-level institution's tendency to seek compromise 805 among competing policy priorities contributed to the error-tolerance of the institutional network. The high-level institution's path-dependent decision-making ensures that the whole system is unlikely to adjust policies drastically. Hence, the incrementalism that derives from the polycentric institutional network structure may help to avoid critical policy failures, which is particularly important in the land system. This could also help to simulate the consequences of widespread 810 distrust between policy actors in large networks (Fischer et al., 2016).

## 4.4 Contextual coherence does not equal logical consistency

While the agents' performance may reflect certain real-world phenomena within institutional 815 networks, it is essential to address a deeper reflection on the current working mechanisms of LLMs. LLMs are designed to optimize literal contextual coherence, meaning that a vast amount of high-quality, textual data enables the machine to effectively mimic human language by approximating patterns of word (or token) changes (Radford et al., 2018). This creates the illusion that LLMs can think. However, while logical reasoning when expressed in a language can lead to contextual 820 coherence, the reverse is not necessarily true. This raises a caveat: over-anthropomorphizing LLM agents can complicate the evaluation of their outputs. This difficulty arises from both the laborious manual work required to assess the agents' logical consistencies and the logical inconsistencies masked by contextual coherence. In future research, LLMs could be trained using "very strict" language that satisfies the condition that contextual coherence leads to logical consistency, which 825 could ensure that the LLM output is logically correct. This would be a significant development for LLM-based simulations.

Understanding the caveat is important for both interpreting LLM outputs and effectively leveraging this approach in modelling human behaviours. LLM agents do not have genuine 830 comprehension of natural language as humans do. This is particularly important when considering the agents dealing with direct and indirect policy influences on the land use system. LLMs are pre-trained on vast amounts of text and excel at recognizing textual patterns. If the policy instruments and targets have strong textual relevance, LLMs should be able to differentiate between direct and indirect policies, just as human policymakers do. Policymakers rely on experience, while LLMs 835 rely on learned associations from their training data. However, effective policymaking is not just

about distinguishing textual relevance. Understanding the real-world impact of policies, including their unintended consequences, is a highly challenging task in complex socio-ecological systems. Given these complexities, current LLMs cannot outperform human policymakers in making policy impact assessments. That said, LLMs can be fine-tuned or prompted to better distinguish direct
and indirect policies, improving their reliability in specific policy contexts (Tao et al., 2024).

## 4.5 Towards broader empirical integration

Although the overall model framework used here is built based on empirical findings, the current
experiments do not capture country-specific policy-making, global interactions or provide sufficient empirical calibration typical of traditional models. The stylized experimental setup – centred on the EU context with a narrowed scope, limited policy targets, instruments, and scenarios – is intended to serve the exploratory and methodological focus. While the LLM agent methodology shows great promise for simulating human-like decision-making, moving toward
fully empirical research necessitates overcoming several significant challenges.

Calibrating socio-environmental models is difficult, and integrating LLM agents introduces additional complicatedness alongside new opportunities. Unlike traditional numerical models with well-defined sensitivity parameters, the performance of text-based LLM agents can be sensitive to
subtle variations in input prompts (He et al., 2024; Mizrahi et al., 2024). Assessing how textual variations in inputs affect the numerical results of the land-use model is inherently challenging, given the vast number of possible word combinations. Robust prompt engineering remains an iterative, experimental process without standardized protocols, complicating the precise alignment of agent outputs with real-world policymaker behaviour. Verifying the correctness of these outputs
is challenging because policy decisions do not have an objectively "correct" answer, which necessitates rigorous manual review by domain experts. These challenges may considerably intensify when factoring in country-specific agents and policies, a necessity within the EU due to its many member states. Moreover, LLM biases should be addressed strategically when applied to empirical research. Multiple factors contribute to biases in LLMs, including prompt engineering,
pre- and post-training processes, and the inherent design of the underlying algorithms (Gallegos et al., 2024). These biases can manifest in various ways, such as political stances (Zhou et al., 2024), cultural perspectives (Liu, 2024), or disparities in responses due to the language in which an LLM is instructed to operate (Tao et al., 2024). However, the primary challenge is not to eliminate biases entirely but rather to shape and align these biases in a controlled manner, so that the LLMs can
sufficiently capture real-world decision-maker behaviours.

To address these challenges, future research could develop quantitative measures for output stability (such as consistency or coherence scores), establish standardized prompt design guidelines, and employ hybrid evaluation frameworks that combine automated metrics with expert
assessments. Despite these hurdles, LLM agents are a rapidly evolving and promising tool that offers a flexible means to simulate multi-actor governance systems, and our work could serve as a foundation that future empirical research can build upon to further refine the methodologies and enhance the representation of human factors in broad socio-environmental modelling.

## 5. Conclusion

We explored the development and application of InsNet-CRAFTY v1.0, a multi-LLM-agent institutional network model with a polycentric structure that is coupled with an agent-based, land system model. By leveraging LLMs to facilitate interactions through textual data, the model enables each modelled entity to pursue unique goals and values that collectively impact the modelled land use system. The results demonstrate that this LLM-enhanced approach is powerful and flexible in modelling institutional actors' behaviours within the land system. However, this novel approach also brings new challenges arising from the limitations of current LLM technology, signifying the need for future research.

*Code and data availability.* All data and code to run InsNet-CRAFTY version 1.0 are made freely available online via Zenodo (https://doi.org/10.5281/zenodo.13944650, Zeng, 2024a; https://doi.org/10.5281/zenodo.13356487, Zeng, 2024b)

*Author contributions.* YZ, CB, and MR contributed to developing the model concept. YZ designed the work, developed the new model and code, and conducted the formal analysis. YZ, CB, JR, and TS wrote the original draft. YZ and MB performed the visualization. CB and MR managed the project administration, while MR acquired funding and supervised the research. All authors reviewed, edited, and validated the final work.

*Competing Interests.* The authors declare that they have no conflict of interest.

*Acknowledgements.* This work was supported by the Helmholtz Excellence Recruiting Initiative, Climate Mitigation and Bioeconomy Pathways for Sustainable Forestry (CLIMB-FOREST; grant no. 101059888) and Co-designing Holistic Forest-based Policy Pathways for Climate Change Mitigation (grant no. 101056755) projects.

## Appendix A

### 1. Terminologies

Table A1: Key terminologies and their definitions/explanations within the context of this research

| Term | Definition |
|---|---|
| Institution | An organization or governing body involved in policy-making, such as government agencies, research institutions, or NGOs. |
| Institutional Dynamics | The interactions, adaptations, and power relations among these institutions over time, which influence how policies are formulated, negotiated, and implemented. |

| | |
|---|---|
| Institutional Network | A structured system of interconnected institutions that interact within the policy-making landscape. |
| Agent | A computational entity within the model that represents an institution, stakeholder, or decision-making body. Agents driven by LLMs in institutional networks can autonomously make decisions, process information, and interact with other agents. Agents in the CRAFTY land use model are rule-based decision-makers, representing various types of land users. |
| Actor | A general term for entities (individuals or organizations) involved in decision-making processes. |
| Long-term/Short-term Memory | Memory components of an LLM-powered agent that influence its decision-making process. Short-term memory refers to immediately available contextual information embedded in the agent's prompt. Long-term memory is stored information retrieved when needed, using data retrieval techniques such as RAG, allowing agents to reference past information and knowledge base. |
| Tools (related to LLM agents) | External functions, APIs, or programming scripts that LLM agents can execute to complete tasks beyond text generation. Here, tools include Python scripts for data analysis, information retrieval, and numerical computations. |
| Tokens | The fundamental units of text processed by an LLM, representing words, characters, or sub-words. Token usage is a consideration in computational cost and efficiency when running LLM-powered agents. LLM providers normally charge API users based on input/output tokens. |
| Context Window | The maximum number of tokens an LLM can process at once. |
| Intermediate Output | Partial results generated by an LLM agent before reaching a final decision. Intermediate output allows agents to iterate on their reasoning, refine calculations, and update their responses before producing a definitive action. |
| Hallucination | The generation of plausible-sounding but factually incorrect or logically flawed responses by an LLM agent. |
| Error | In a computational model, an "error" typically refers to a deviation from the intended functionality or an outcome that violates the system's logical rules or design specifications. |
| Erroneous Behaviour | We broadly defined errors related to the LLM agents in terms of logical coherence, contextual awareness, or workflows as erroneous behaviours. In this paper, erroneous behaviour includes misinterpretations of data, incorrect formatting, logical inconsistencies, contextual incoherence, or failures in workflow execution. |
| Counter-intuitive Behaviour | Counter-intuitive behaviour refers to an emergent pattern or decision-making outcome that deviates from conventional expectations or common sense. Counter-intuitive behaviour occurs when agent decisions do not align with typical policy-making norms, such as the unexpected budget allocation, which may contradict the assumption that the percentage of the total budget should be 100%. This behaviour does not necessarily indicate an error but rather an unanticipated outcome driven by agent decision-making. |
| Believable Behaviour | Believable behaviour means that an agent's actions and decision-making processes resemble realistic human behaviour, or at least capture some crucial aspects of human activities. |
| Robustness | Robustness refers to the ability of the model to function effectively despite errors, incomplete information, or unexpected disruptions. A robust system maintains functionality even when agents produce erroneous outputs or misinterpret data. This type of robustness does not imply statistical robustness (which requires sensitivity |

| | analysis) but rather operational resilience, meaning that the model does not collapse or produce entirely unrealistic behaviours due to minor errors. |
|---|---|

## 2. Model Processes

Figure A1 illustrates the model processes, segmented into three distinct sections. The land use
section (blue) encompasses all processes directly related to changes in land use. The LLM agent
section (green) consists of the activities performed by LLM agents. The operational institution is
a hybrid agent, integrating rule-based decision-making processes (yellow) and LLM-driven
procedures (procedures 4 and 21).

This hybrid approach aligns with the dual nature of organizational routines and non-routine actions,
as extensively analysed by Simon in his seminal work, *Administrative Behavior* (Simon, 2013).
Organizational routines are recurring actions embedded in an organization's culture, ensuring
consistency and efficiency. In contrast, non-routine actions are spontaneous and designed to
address unique, unpredictable situations. Both are crucial for effective organizational functioning.
The rule-based components correspond to organizational routines, ensuring strict adherence to
930 operational protocols, while the LLM component allows for creative, sometimes imperfect,
responses.

In InsNet-CRAFTY, the LLM-related functionalities of the agents are written in Python, while the
rule-based processes and CRAFTY are coded in Java. The sub-models written in the two
programming languages are connected through a client-server architecture. For a comprehensive
description of the rule-based processes within the operational institution (steps 6-14), refer to Zeng
et al. (2025b). The explanations of the processes within each section are provided below.

**Steps 1 – 4:** Launching the server end and initialising the LLM agents. This includes creating a
940 server object that listens to requests from the client end and instantiating the agent class by
initializing the model names of the large language models, API keys, prompt templates of the LLM
agents, and agent-specific workflows. The optional narrative injector is not displayed here.

**Step 5**: Launching the client end and initialising the CRAFTY land use model. Key procedures
include initializing the distribution of capitals and agent functional types, i.e., AFTs (Brown et al.,
2019; Murray-Rust et al., 2014).

**Steps 6 – 14:** Rule-based policy adaptation of the operational institutions. Step 6 includes the
initialization of the operational institutions' rule-based components, initial policy goals and
950 accessible policy instruments. At step 7, each operational institution collects information from the
land use model. Step 8 determines if it is time to trigger the policy adaptation processes. If Step 8
outputs "true", the operational institution starts to evaluate the current policy's performance using
a PID (Proportional-Integral-Derivative) mechanism and calculate the needed policy intervention
intensity using a fuzzy logic controller (Zeng et al., 2025b), which can convert experts' knowledge
into computer-comprehensible rules to automate the decision-making (step 9 and 10). Step 11 is a
normalised non-monetary constraint restricting the policy change. Steps 12 and 13 further tailor
the policy change to satisfy the budgetary constraints. Step 14 is the resultant policy adaptation.

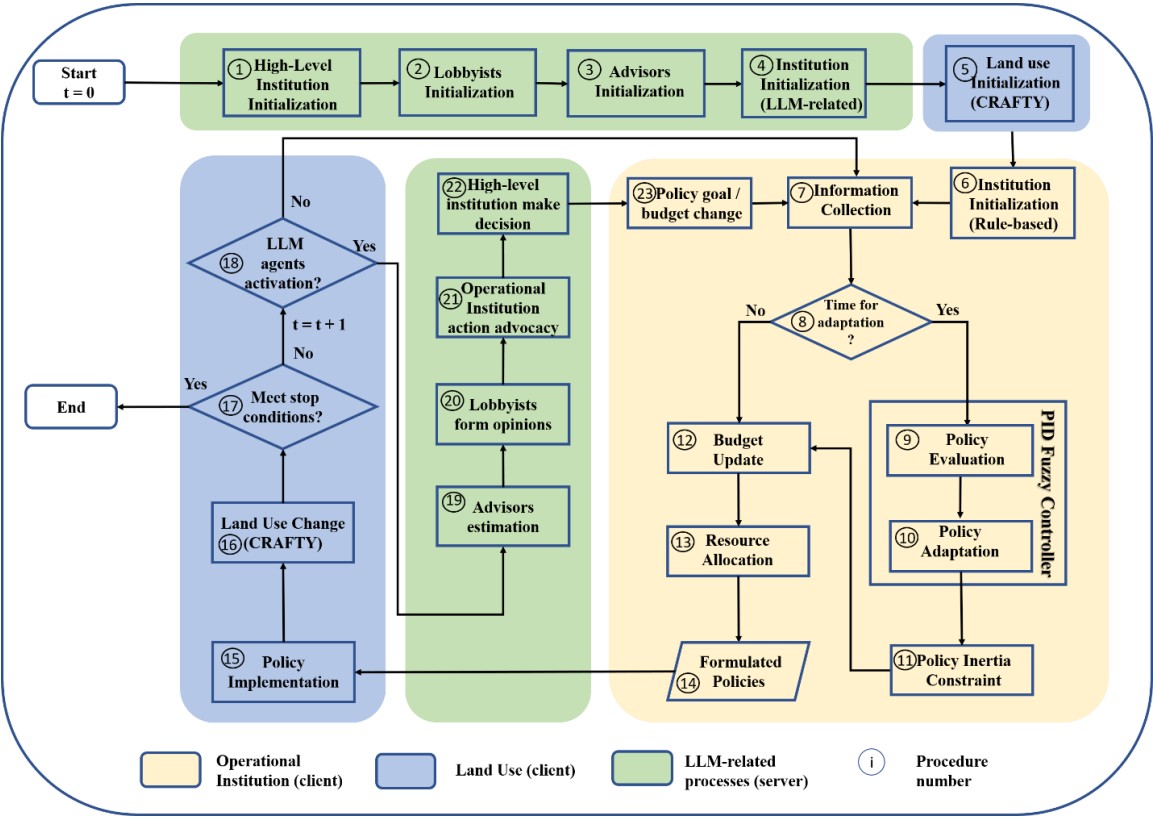

Figure A1: Model processes of InsNet-CRAFTY v1.0

**Step 15**: Policy implementation. Implement the policy by changing corresponding variables in the land use model.

**Step 16**: Land use change updating. Run the land use model under the influence of policy interventions. This produces responses of the land use model in terms of land use type distributions and ecosystem services' demand and supply.

**Step 17**: Termination check. Check if it is time to terminate the whole simulation.

**Step 18**: LLM interaction check. Check if it is time to trigger the LLM agents. If false, go back to step 7; otherwise, go to step 19 by sending a request and the updated land use data to the server for the policy goals as well as budget allocation formulated through the LLM agents.

**Step 19 – 22**: LLM agent activation. Activate the LLM agents on the server end to obtain the output of each of them. The narrative injector outputs the updated narratives (optional); the research supplier provides the textual interpretation of the numerical results collected from the land use model; The lobbyists construct their arguments for their benefits; the operational institution's LLM module can also generate arguments to propose financial support and proper policy goal adjustments; the high-level institution receives all the information to form the final

decision in terms of budget allocation and policy goal adjustments, which in turn influence the behaviour of the operational institutions.

**Step 23**: Sending back the updated policy goals and budget allocation to the operational institutions, based on which the operational institutions adjusted their policy-making.

# Appendix B

Table B1: Prompt template of the research supplier

You are an AI assistant in policy-making that can write Python code to analyze data.
You are responsible for debugging your own code using the provided tools.
Now, analyse the data in the CSV file in the way you think appropriate.
You can reference the following instructions to conduct your analysis step by step.
Step-by-step data analysis instructions:
1. **Load the Data**
  - Load the CSV file into a DataFrame.
  - Display the first few rows of the DataFrame to understand the structure and the types of data included.
  - Check for missing values or inconsistencies in the data.
2. **Initial Data Inspection**
  - Use descriptive statistics (like `data.describe()`) to get an overview of the numerical features.
  - Plot histograms or box plots for each numerical feature to understand the distribution and spot any outliers.
3. **Detailed Analysis of Specific Features**
  - **Meat Production Analysis**:
   - Plot time-series graphs for meat demand, meat supply, and policy goals for meat production.
   - Analyze the trends and gaps between demand, supply, and policy goals.
  - **Protected Area Analysis**:
   - Plot the protected area ratio over time alongside the policy goals for the protected area ratio.
   - Identify any discrepancies between policy goals and actual outcomes.
4. **Budget Allocation Analysis**
  - Create line plots to visualize the budget allocations for meat production and protected areas over time.
  - Compare these allocations to see if the budget is aligned with the goals and outcomes.
5. **Evaluate Correlations and Causations**
  - Investigate correlations between different variables using scatter plots or correlation matrices.
  - Consider potential causative factors that could explain trends observed in the data.
6. **Summarize Findings**
  - Summarize key insights into specific bullet points from the data analysis.

Table B2: The prompt template of the environmental NGO

You are a representative of an environmental NGO that is concerned with environmental protection and climate change.
Based on the information given below and the identity of your role, generate some bullet points to lobby the high-level public policy institution to prioritise nature conservation.
The given information: {given_information}

Table B3: Prompt template of the land user association. A specific role – a representative of the meat production industry – is given to the agent to enable it to focus on a concrete topic.

You are a representative of the meat production industry, who cares about the benefits of the industry.
Based on the information given below and the identity of your role, generate some bullet points to lobby the high-level public policy institution to prioritise meat industry development.
The given information: {given_information}

Table B4: The prompt template of the agricultural institution

As a policy-maker specializing in agriculture, you oversee initiatives critical to your region's food security, farmer livelihood, and financial well-being.
Currently, you're focusing on meat production, a sector facing significant challenges due to changing market demands.
Your role is to propose a set of compelling and concise bullet points to the high-level institution, seeking increased priorities and financial support for meat production.
Consider the economic impact and social implications. Specifically, you should prompt the high-level institution to make reasonable policy goals that align with budget allocation.
Use the data in the CSV file provided to argue your case effectively.

Table B5: The prompt template of the environmental institution

As a policy-maker specializing in environmental protection, you oversee initiatives critical to nature conservation, biodiversity, and pollution reduction including the Net-zero targets in your region.
Currently, you're focusing on the expansion of protected areas, a sector facing significant challenges due to biodiversity loss.
Your role is to propose a set of compelling and concise bullet points to the high-level institution, seeking increased priorities and financial support for protected area establishment. Specifically, you should prompt the high-level institution to make reasonable policy goals and budget allocation.
Use the data in the CSV file provided to argue your case effectively.

Table B6: The prompt template of the law consultant

You are a law consultant giving advice to a high-level public policy institution that is responsible for making public policies regarding agricultural production and environmental protection.
Use the provided context about relevant policies, laws, regulations, etc., only to form your advice to ensure the high-level institution makes policies legally.
(if you don't know the answer in the given context, just say you don't know):
    <context>
    {context}
    </context>
Question: {input}

Table B7: Prompt template of the high-level institution

Simulation Role: You are a high-ranking policymaker in charge of overseeing operational institutions within the land system.
Key Actions:
Budget Allocation: Allocate the financial resources between the Agricultural and Environmental Institutions. This directly affects their operational capabilities and initiatives.
Policy Goal Adjustment: Adjusting policy goals appropriately for each institution.
Objective:
Strategically guiding operational institutions, including Agricultural and Environmental Institutions; harmoniously balancing the interests of diverse stakeholders.
Input information:
1) Input from Agricultural institution: {AgriInstInput}
2) Input from Environmental institution: {EnviInstInput}
3) Input from Environmental NGO: {NGOInput}
4) Input from Land user association: {landUserInput}
5) Input from the environment: {narrIput}
6) Input from research suppliers: {researchInput}
7) Historical information: {history}
8) Law consultant: {lawInfo}
Decision-Making Guidance:
Be explicit about your role as a policymaker and your impact on operational institutions.
Make informed decisions by thoroughly analyzing inputs from all stakeholders.
Reflect on historical information to inform decisions.
Ensure that your actions and decisions are logical, well-reasoned, and transparent.
Before giving your final decision, provide a step-by-step rationale for each decision, showing how it aligns to balance stakeholder interests and ensure the feasibility of policy adjustments.
The rationale should support you in quantifying the planned changes in each operational institution's budget and policy goal using percentages.
Note:

The long-term goals have already been specified, your tasks are dynamically conducting reasonable modifications to the goals and providing feasible budget allocation to support the achievement of the goals.
Output requirements:
1. Output your step-by-step reasoning here: including stakeholder input analysis, budget allocation analysis, and policy goal adjustment analysis.
2. Format your quantified policy adjustments using JSON. Your output should be a clean JSON without anything beyond. An example is as follows:
{
   "Budget Allocation": {
      "Agricultural Institution": using a positive integer to indicate the percentage of budget allocation here,
      "Environmental Institution": using a positive integer to indicate the percentage of budget allocation here
   },
   "Policy Goal": {
      "Agricultural Institution": using an integer to indicate the percentage of policy goal change here; positive integers indicate the percentage of increase in the current policy goal, while negative ones mean decreasing the current policy goal; 0 means remaining the current policy goal unchanged,
      "Environmental Institution": using an integer to indicate the percentage of policy goal change here; positive integers indicate the percentage of increase in the current policy goal, while negative ones mean decreasing the current policy goal; 0 means remaining the current policy goal unchanged
   }
}

1010

# Appendix C

One can consider the high-level institution as a controller over the operational institutions, which in turn impose their control over the land use model. As previously stated, the operational institutional agents are hybrid. They incorporate both a LLM component to interact with other LLM agents and rule-based behaviour to interact with the programmed land use model. We use the endogenous institutional model described in Zeng et al. (2025b) to simulate the rule-based behaviour of the operational institutional agents. We first describe how the operational institutions' non-LLM modules work, and then introduce how the high-level institution's influence comes into play.

## 1. Operational institution

### 1.1 Policy goal definition

The first step to model an operational institution's behaviour is to define a policy goal, which can be represented by a three-dimensional vector:

$$\mathbf{G^i} = \left[T_s^i, T_e^i, Q^i\right] \tag{C1}$$

meaning operational institution $i$'s policy goal consists of $T_s^i$ the time when the policy starts, $T_e^i$ the time when the policy ends, and $Q^i$ the quantity of an ecosystem outcome a policy is meant to change during the time from $T_s^i$ to $T_e^i$. For instance, if we only have two operational institutions, e.g., an environmental institution and an agricultural institution, the possible values of i can only be 1 or 2.

## 1.2 Policy evaluation and adaptation

The operational institutions estimate their policy effectiveness using Eq. (C2):

$$E_{t_n} = \frac{1}{k} \sum_{m=n-k}^{n} \frac{Q^i - o_{t_m}^i}{|Q^i|} \tag{C2}$$

where $t_n$ represents the specific time at which the institution evaluates the goal-outcome error $E_{t_n}$; $o_{t_m}^i$ is the outcome intended to be adjusted by institution $i$ at the time $t_m$; $k$ is the time interval of interest.

Let $F$ denote the function of a fuzzy logic controller (FLC) and $F(E)$ indicate policy variation. The constrained policy variation $A_{t+1}^i$ at $t+1$ is calculated as

$$A_{t+1}^i = sign\big(F(E)\big) \times \min\big(|F(E)|, N^i\big) \tag{C3}$$

The above equation means that the absolute value of policy variation within one iteration should be no greater than policy inertia constraint $N^i$. The sign function outputs the sign (+1, 0, or -1) of its input.

$A_{t+1}^i$ is accumulated to form a policy modifier denoted as $M_{t+1}^i$, as shown in Eq. (C4).

$$M_{t+1}^i = M_t^i + A_{t+1}^i \tag{C4}$$

The policy variation is normalised and used with a fixed step size for iterative policy adaptation. The policy modifier is a coefficient of the step size. As shown in Eq. (C5), $\eta^i$ is the step size, and $V_{t+1}^i$ is the modified policy intervention for the $(t+1)$-th iteration.

$$V_{t+1}^i = \eta^i \times M_{t+1}^i \tag{C5}$$

The budget update process monitors the institution's income and expenditure whenever a policy is implemented. This assumes that policy interventions can be quantitatively measured, with their absolute values being positively correlated with the budget required by the institution to implement the policy. In Eq. (C6), $f$ represents a monotonic function that maps the absolute value of a policy intervention $V_{t+1}^i$ to resource $R_{t+1}^i$ needed to carry out this policy. In this model, only subsidization

and the establishment of new protected areas require budget allocations; the costs associated with taxation are not included.

$$R^i_{t+1} = f\left(\left|V^i_{t+1}\right|\right) \tag{C6}$$

The actual policy intervention under the budgetary constraint is

$$V^i \leftarrow \text{sign}(V^i) \times f^{-1}\left(min\left(R^i_{t+1}, B^i\right)\right) \tag{C7}$$

The budget of operational institution i should be updated via operation (C8):

$$B^i \leftarrow \max(B^i - R^i_{t+1}, 0) \tag{C8}$$

The implemented policies are supposed to influence land users' behaviour. In CRAFTY (Murray-Rust et al., 2014), land users are categorized into an array of AFTs (Agent Functional Types), each of which can provide multiple ecosystem services. AFTs differ in their capabilities of using a diversity of capitals within land. The AFTs compete for land in the pursuit of benefit, which in turn influences the whole system's ecosystem service supply.

## 1.3 Policy implementation

In a rasterized map, the competitiveness of an AFT under the influence of economic policies (such as subsidies and taxes) can be calculated as follows:

$$c_{xy} = \sum_S (p_S \left(\sum_i V^{iS}_{ECON} + m_S\right)) \tag{C9}$$

where $c_{xy}$ denotes the competitiveness of a land use agent at the land cell $(x, y)$; $S$ is the ecosystem service the land user produces; $p_S$ is the total production of S within the land cell; $V^{iS}_{ECON}$ is the institution $i$'s economic policy that targets ecosystem service $S$; $m_S$ is marginal utility brought by ecosystem service $S$.

The environmental institution identifies the top N unprotected land cells within the model based on the richness of a chosen set of capitals requiring conservation. Here, two natural capitals defined in the CRAFTY-EU (Brown et al., 2019), i.e., forest and grassland productivity, are used to determine if a land cell needs protection. The value of N at each stage is determined using the previously mentioned fuzzy controller method. Typically, if there is a significant gap between the PA target and the current PA coverage, the value of N would be increased. Certain products cannot be produced on the protected land cells. Therefore, the competitiveness of an AFT on protected land cells can be calculated as:

$$c_{xy} = \sum_S (w_S p_S \left(\sum_i V^{iS}_{t+1} + m_S\right)) \tag{C10}$$

where $w_S$ represents an element of a vector $w$ whose elements equal either one or zero, which defines whether a type of ecosystem service is allowed to be produced in PAs (Zeng et al., 2025a).

The CRAFTY-EU model considers seven types of ecosystem service (including meat, crops, habitat diversity, timber, carbon, urban, and recreation). In the current model setting, it is assumed that only habitat diversity is allowed to be improved by the AFTs in the PAs, reflecting a strict restriction on ecosystem service production.

## 2. High-level institution

To let the model form a self-sustained system, it is assumed that the total budget obtained by the high-level institution is related to the total production of the ecosystem services, corresponding to the fact that governmental incomes are mainly from the gross domestic product (GDP).

$$B_t^{total} = \alpha \sum_S P_{S,t} \tag{C11}$$

where $B_t^{total}$ means the total budget the high-level institution can allocate between the operational institutions at t; $P_{S,t}$ represents the total production of ecosystem service S across all AFTs at time t; $\alpha$ is a coefficient that indicates the proportion of the total budget to total ecosystem service production.

The budget gain $\triangle b_{i,t}$ of operational institution i at time t is calculated as

$$\triangle b_{i,t} = \beta_i B_t^{total} \tag{C12}$$

where $\beta_i$ is the percentage controlled by the high-level institution. Hence, the budget of operational institution i should be updated:

$$B^i \leftarrow B^i + \triangle b_{i,t} \tag{C13}$$

Whenever the high-level institution adjusts operational institution i's policy goal by a percentage $\triangle q^i$, the policy goal is updated as

$$Q^i \leftarrow Q^i(1 + \triangle q^i) \tag{C14}$$

Operation (C7) indicates that operational institutions cannot consume resources more than their budget. However, the equation does imply that the budget can be insufficient for implementing a policy. We use the difference between operational institution i's budget and the needed resources to calculate the budget surplus at time t using Eq. (C15). Therefore, the budget surplus can be either positive or negative.

$$SUR_{i,t} = B^i - R_t^i \tag{C15}$$

## 3. Numerical settings

Tables C1 – C3 show the numerical settings related to the policy-making processes of the operational institutions.

Table C1: The settings of the operational institutions and the high-level institution

| Institution attributes | Settings |
|---|---|
| Unique ID | "Agricultural_Institution" |
| Policies | "Meat_economic" |
| Information | Annual meat supply and demand, budget surplus. |
| Budget | Allocated by the high-level institution based on total ecosystem service production annually. |
| Decision rules | "Economic" |

| **Policy attributes** | |
|---|---|
| Unique ID | "Meat_economic" |
| Target service | "Meat" |
| Policy Type | "Economic" (see Table C2) |
| Step size $\eta^1$ | 1000000 |
| Inertia constraint | 1.0 |
| Initial policy goal | 120% initial meat production |
| Time lag | 2 |
| Policy-resource function | $R = f(|V|) = max(V, 0)$<br>(Note: Only if V > 0, does the institution consume budget, and the budget use equals the subsidy.) |

| **Institution attributes** | Settings |
|---|---|
| Unique ID | "Environmental_Institution" |
| Policies | "Protected_areas" |
| Information | Protected area ratio |
| Budget | Allocated by the high-level institution based on total ecosystem service production annually. |
| Decision rules | "Protection" (see Table C3) |

| **Policy attributes** | |
|---|---|
| Unique ID | "Protected_areas" |
| Target service | "Protected areas" |
| Policy Type | "Protection" |
| Step size $\eta^2$ | 1.0 |
| Initial policy goal | 10% of total land |
| Initial guess | 10000 |
| Time lag | 2 |
| Timer | Equal to the time lag |
| Adapting | False |
| Policy-resource function | $R = f(|V|) = 1000V$<br>(Note: V indicates the number of land cells that need to be protected, and it is assumed that each new protected cell |

| Institution attributes | Settings |
|---|---|
| | consumes 1000 units of budget. The value is set for making the budget consumptions of the two operational institutions comparable.) |
| α | 0.01<br>(Note: The high-level institution uses 0.01 times the total ecosystem production of the modelled system as the total budget that can be allocated between the two operational institutions) |

1115

Table C2: Parameterisation of the fuzzy decision rules labelled as "Economic", using FLC language defined in the IEC 61131-7 (2024)

```
FUNCTION_BLOCK Economic
VAR_INPUT
        gap: REAL;
END_VAR
VAR_OUTPUT
        Intervention : REAL;
END_VAR
FUZZIFY gap
        TERM nhigh := (-0.5,1) (-0.3,0);
        TERM nmild := (-0.5,0) (-0.3,1) (-0.1,0);
        TERM nlight := (-0.3,0) (-0.1,1) (0,0);
        TERM neutral := (-0.05,0) (0,1) (0.05,0);
        TERM plight := (0, 0) (0.1, 1) (0.3,0);
        TERM pmild := (0.1,0) (0.3,1) (0.5,0);
        TERM phigh := (0.3, 0) (0.5, 1);
END_FUZZIFY
DEFUZZIFY intervention
        TERM nhigh := (-0.2,1) (-0.1,0);
        TERM nmild := (-0.15,0) (-0.05,1) (0,0);
        TERM neutral := (-0.02,0) (0,1) (0.02,0);
        TERM pmild := (0,0) (0.05,1) (0.15,0);
        TERM phigh := (0.1,0) (0.2,1);

        METHOD : COG;
        DEFAULT := 0;
END_DEFUZZIFY
RULEBLOCK No1
        AND : MIN;
        ACT : MIN;
        ACCU : MAX;

        RULE 1 : IF gap IS nhigh  THEN intervention IS nhigh;
        RULE 2 : IF gap IS nmild  THEN intervention IS nmild;
        RULE 3 : IF gap IS nlight THEN intervention IS neutral;
        RULE 4 : IF gap IS neutral THEN intervention IS neutral;
        RULE 5 : IF gap IS plight THEN intervention IS neutral;
```

```
        RULE 6 : IF gap IS pmild THEN intervention IS pmild;
        RULE 7 : IF gap IS phigh THEN intervention IS phigh;
END_RULEBLOCK
END_FUNCTION_BLOCK
```

1120

Table C3: Parameterisation of fuzzy decision rules labelled as "Protection", FLC language
defined in the IEC 61131-7 (2024)

```
FUNCTION_BLOCK Protection
VAR_INPUT
        gap: REAL;
END_VAR
VAR_OUTPUT
        intervention : REAL;
END_VAR
FUZZIFY gap
        TERM plow := (0,1) (0.15,0);
        TERM plight := (0.025, 0) (0.175, 1) (0.325,0);
        TERM pmild := (0.175,0) (0.325,1) (0.45,0);
        TERM phigh := (0.325, 0) (0.45, 1);
END_FUZZIFY
DEFUZZIFY intervention
        TERM neutral := (0,1) (0.075,0);
        TERM plight := (0.025,0) (0.075,1) (0.125,0);
        TERM pmild := (0.075,0) (0.125,1) (0.175,0);
        TERM phigh := (0.125,0) (0.2,1);

        METHOD : COG;
        DEFAULT := 0;
END_DEFUZZIFY
RULEBLOCK No1
        AND : MIN;
        ACT : MIN;
        ACCU : MAX;

        RULE 0 : IF gap IS plow THEN intervention IS neutral;
        RULE 1 : IF gap IS plight THEN intervention IS plight;
        RULE 2 : IF gap IS pmild THEN intervention IS pmild;
        RULE 3 : IF gap IS phigh THEN intervention IS phigh;
END_RULEBLOCK
END_FUNCTION_BLOCK
```

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
