# Peer review of "InsNet-CRAFTY v1.0: Integrating institutional network dynamics powered by large language models with land use change simulation"

_EGUsphere, 2024_

## Referee Comment (RC1)

[referee-annotated manuscript omitted]

---

## Author Comment (AC1)

**Reply on RC1**

- **Black text** indicates reviewer comments.
- **Blue text** indicates author responses.

**Summary of the review**

Zeng et al. developed an innovative LLM model that simulates interactions between institutional agents that can mimic reasoning, planning and action. The model is novel because it addresses key challenges that learning and memory and polycentricity and because it is linked to an agent-based model that simulates changes in land use and livelihoods. The development of the LLM model described in the paper is ambitious and challenging. Certainly, it cannot be expected that all issues and challenges are yet addressed and that it completely functions as intended. The authors describe the challenges they occur and how they may solve them. It is impressive that the authors make sure that everything becomes available open access.

I read the paper with pleasure. It is generally well-written, novel and informative. However, there are a number of things that need improving in my opinion. That is why I recommend major revisions. In the attached pdf file, the authors can find detailed comments. Below they can find a summary of the issues that are in, my opinion, important to address.

The experimental set up:

The intention of the paper is to test the model and simulate institutional actor's behavior in the land system. Many different types of policy goals can be tested and different types of actors with different types of profiles and ways in which they interact can be chosen. The choices made in the experimental set up affect the outcome of the experiment. At the moment, limited rationale is provided for the experimental set up by the authors. There is no rationale provided for the choice of the SSP-RCP scenario. Additionally, limited rationale is provided for the choice of starting conditions and the types of policies that are considered. Limited rationale is provided for the choice of the combination of agents in the experiment. As this all influences the outcomes of the experiment, it is important that such rationales are provided. There is limited rationale provided for focusing only on the response of institutional agents to EU land system dynamics, without considering effects other regions in the world may have had on the results. Additionally, it is important to discuss how the experimental set up could have influenced the outcomes in unintended ways and which limitations of the model could have been accidentally missed because they did not come into play because of the way the experiment was set up. It would be great if the authors could address this thoroughly in the paper, so that the value of doing this particular experiment, but also its limitations, becomes clearer. At the moment, it was difficult for me to judge if the model is sufficiently tested using through this one experiment to run other types of scenarios with other policy targets, other institutional agents etc. Or whether more tests and sensitivity analyses are necessary for the model to be used more broadly. Especially since the outcomes of budget surplus for PAs and budget deficit for agriculture are a bit counter-intuitive in my perception and seem to reveal an overreliance of agents on policy documents.

Errors and robustness:

The authors speak of error proneness, error tolerance and robustness but these terms are not defined and the process of testing for this is not explained in the methodology. Usually, these terms are used in modelling literature in the context of quantitative sensitivity analysis but here they are used to refer to some unexpected or undesirable behavior of institutional agents. I find this personally a bit confusing, as I do not see so well how the error and robustness of the model could be derived from a qualitative assessment of the agents' behavioral patterns. Therefore, I would recommend to either use different terms, such as undesired or unexpected agent behavior or to really well define the terms around error well and thoroughly describe in the methodology how the authors assessed the errors. If the authors would really like to emphasize error proneness and robustness in the more traditional modelling sense, I would recommend the authors to do additional analyses. For example, to add a sensitivity analysis with different starting conditions, different environmental and social goals or different combinations of agents with different profiles, etc.

Information lacking to interpret results well:

As the LLM model is linked to CRAFTY, it is of course not possible to describe every dynamic of both models in detail in this paper. However, to understand the results and discussion some fundamental modelling assumptions were missing from the main paper, such as the way the budget is modelled and how the agents, for example know how much budget is needed etc. It would be great if the authors could provide more detailed descriptions of such assumptions and modelling choices, so that the results can be more easily interpreted. Or to very explicitly refer the appendices that are adjusted in such a way that the reader can understand the results and their interpretation easily after reading them.

Writing:

Although the model is intended to mimic real-life situations, the description of the model and the results, as well as the discussion of the results remains very high-level and abstract. I would highly recommend including real-life examples of agents in the context of the EU and to discuss the results in context of dynamics at play in the EU. This would all make it a bit more tangible. In particular I would recommend including a discussion of the outcome of the model in context of what happened in the EU in the past and what has been found in previous studies.

Writing style:

The paper is generally well-written. Yet, in some parts of the paper jargon is used and quite some terms that would be up for interpretation remain undefined. It would be good to more specifically define some of the terms, so that the model and results are easier to interpret by readers of different disciplines. This is important because the model can be used in interdisciplinary settings and, when linked to other models, such as CRAFTY can influence land use modelling, which is a different field again altogether. I have put comments throughout the paper that are hopefully helpful to address this.

Dear Reviewer,

Thank you very much for your thorough and detailed review of our manuscript. We greatly appreciate the time and effort you have invested in providing your feedback. For ease of reviewing and revision, we have categorized and numbered your comments. Below, we provide a general overview of your comments and our corresponding approach, with detailed point-bypoint responses following in the subsequent pages. We have addressed many of the comments in the revised manuscript and will upload the completed revision upon Editor Müller's approval. The manuscript with your numbered comments is attached to the end of the responses.

- **Definitions and Terminologies:** We appreciate you highlighted the importance of providing clear definitions and conceptual clarity, especially as this research spans multiple disciplines, which may pose challenges for readers' comprehension if the terms are not adequately explained. We will make every effort to improve clarity through additional explanations and illustrative examples where appropriate, including adding a glossary in the appendix if necessary.
- **Additional Examples:** We recognize the importance of concrete examples in enhancing reader comprehension. We will incorporate additional examples where feasible to further support our arguments.
- **Rephrasing Suggestions:** We appreciate your suggestions for improving readability and will incorporate them accordingly.
- **Text Reorganization:** We will carefully consider your recommendations regarding the restructuring of text and implement changes where they improve logical flow and clarity.
- **Model's Working Mechanisms:** Many of your comments pertain to details provided in the appendices. While we attempted to move some of this information into the main text, we found that doing so often led to redundancy or a loss of contextual coherence due to the interdependency of the equations. To address your concerns, we have followed your suggestion to explicitly reference the relevant appendices wherever needed in the main text to ensure clearer guidance for readers.
- **Experimental Settings:** As noted, our model builds upon previous research, and many of the land use model settings have been documented in prior publications. We will strive to strike a balance between providing sufficient detail and avoiding unnecessary repetition. Additionally, we would like to mention that our framework is directly derived from an empirical study of the EU political system. However, it is intentionally designed to be relatively abstract and stylized, as the current experiments mainly serve proof-of-concept and exploratory purposes. The experimental settings of the AI agents are currently hypothetical and designed to be simple and straightforward. Rather than aiming to derive operable policies for real-world implementation, the primary focus is on investigating the internal logic coherence and contextual awareness of the AI agents, as well as exploring their potential integration with existing rule-based land use models.
- **Punctuation and Typographical Issues:** We greatly appreciate your meticulous attention to detail. We will carefully review and correct all punctuation and typographical errors.
- **Others:** We acknowledge these and will strive to address them carefully to improve the overall quality of the manuscript.

Sincerely,

Yongchao Zeng

On behalf of all authors

**Page 1:**

1. Maybe revise the first sentence. Understanding and modeling environmental policy interventions does not directly contribute to land use and management. Perhaps you can turn it around and state something like: To foster sustainable land use and management an increased understanding of ways in which policy interventions can contribute etc. etc. Modeling such processes can help derive this understanding.
Good suggestions. This sentence has been improved in the revised manuscript.

2. ...that, while...
Done.

3. This term seems a bit vague to me, the tolerance part I mean (but perhaps it's jargon from the field I don't know). Could you state something like that the network is robust to.... Or just low error or something.
Thanks. The sentence has been rephrased.

4. Very long sentence, quite difficult to follow.
Done.

**Page 2:**

5.   ...topics, such as...
Done.

6.   Reference missing.
Reference added.

7. Could you give an example here of actors and how they relate to each other (e.g., further building upon the EU example)?
The formation and implementation of land use policies are the product of complex institutional dynamics and can involve a wide range of actors with differing objectives and powers (Davidson et al., 2024). For instance, within the European Union (EU), multi-level governance systems play a significant role. At the EU level, institutions like the European Commission propose policies such as the Common Agricultural Policy (CAP), which sets broad objectives for sustainable land management, rural development and food security. These policies are then negotiated with the European Parliament and the EU Council, which consists of ministers from each EU country. National governments, in turn, are responsible for tailoring these policies to their domestic contexts, often collaborating with regional governments, local municipalities, and non-governmental organizations (NGOs). For example, an NGO advocating for biodiversity conservation might work alongside local authorities to implement EU directives, such as the Natura 2000 framework, ensuring compliance with broader policy objectives while addressing local needs. See Box 1 (at the end of the responses) for more details.

8. Could you define institutions and/or institutional dynamics? In other disciplines institutions can mean a range of things

Here we define institutions as organizations or governing bodies involved in policy-making, such as government agencies, research institutions, or NGOs. Institutional dynamics encompass the interactions, adaptations, and power relations among these institutions over time, which influence how policies are formulated, negotiated, and implemented.

We will add the above definitions to the manuscript.

9. It feels to me a bit like jumping to a conclusion. Could you explain a bit before how actors can shape land use systems and how their interactions and the influence this has on policies matters? Maybe using the example on the EU?

Referring to the CAP as an example once again, national and local government implementation can vary based on economic, social and environmental priorities. Local actors, including farmers and regional governments, further influence land use through on-the-ground practices and lobbying efforts. These interactions—whether cooperative or contentious—can result in policy outcomes that either advance or hinder environmental goals. For instance, tensions between biodiversity conservation objectives and agricultural production have led to debates over subsidies and land management practices, demonstrating how institutional dynamics can shape the land system and its environmental implications.

10. Would it be possible to move this paragraph after the next one (so after the paragraph that's currently the third one)? I think it is easier perhaps to understand what it requires to have a holistic representation of institutional actors. Hereafter, this second paragraph can provide information on how things were done before and then moving on to the LLMs. By doing this you likely also regard some of the comments I provided before and after this paragraph.

Thank you for the suggestion. We would prefer to keep the order as is, with the proposed additions to the paragraphs. As such, we would first state the current research to identify gaps, and then propose new approaches (i.e., the LLM method) to address these gaps.

11. The importance is not really well described. Please refer to my previous comments. In my opinion there is a bit more context and some examples needed to make this point.

Thanks, and yes, we believe the additional context and examples will allow us to better describe the importance here.

12. How is this defined?

We have added the institutions and institutional dynamics definitions above, in response to Comment 8. We will also rephrase the text to "networks of institutions" so it is clearer what we are referring to.

13. Better not to refer to previous paragraph like this at the start of a new one.

Thanks, the sentence will be revised accordingly.

14. Maybe better to not yet use "we" at this point in the introduction.

We will revise for consistent usage throughout.

**Page 3:**

15....model, InsNet-CRAFTY, and...

Done.

16. Adopted
Done.

**Page 4:**

17. Potentially two spaces here between maintains and Gonzalez.
Thanks, we have corrected it.

18. I am a bit confused about the terms "agents" and "actors". It seems you use these terms interchangeably in some cases but in other cases they seem to be used in a distinct way (e.g., in case of CRAFTY agents). Could you define these two terms and be consistent?

Yes, we will check for consistency.

"Agent" means a computational entity within the model that represents an institution, stakeholder, or decision-making body. Agents in the institutional network are powered by LLMs and autonomously make decisions, process information, and interact with other agents. Agents in the CRAFTY land use model represent various types of land users.

"Actor" is a general term for entities (individuals or organizations) involved in decision-making processes. Actors can be both real-world stakeholders (e.g., policy-makers, lobbyists) and their simulated counterparts (LLM-powered agents).

**Page 5:**

19. Could you give an example of a high-level institution in real life? And could you do the same for all other actors and agents? This would make things a bit more tangible and less abstract.
A good suggestion, thanks - we will do so. The level here is a relative term, and so real-world institutions can be at any level depending on the context being considered. We only need to know whether the role is relatively higher in the structure, and it is not the front-line institutions that execute specific policies. For example, the high-level institution can be the European Commission, which is a supranational policymaking entity. The high-level institution sets the overall policy ambitions and constraints (e.g., budgets) that affect the decisions of the Member States. Within the EU, the European Commission develops and proposes policy frameworks, such as the European Green Deal, aiming to achieve long-term goals like climate neutrality by 2050, based on the information provided by the operational institutions, research suppliers, lobbyists, and law consultants. For more details, please see Box 1.

20. Could you be more specific and/or give an example of this data?

We will specify here yes, by referring to the description of the data in the experiment setting section, i.e., "the data used here is described in detail in section 2.3".

**Page 6:**

21. I see you are still providing the example here of PAs and meat supply. I didn't realize up to this point that you were still providing an example. Could you make it more clear? I suddenly thought this was a specific focus or the only thing that could be modeled.
We will make it clearer in the first mention of meat and PAs that this is just an example. We will also modify the text to explain the importance of this example:

"For instance, CRAFTY can produce information indicating that the supply of certain food products (e.g. meat) and the coverage of protected areas (PAs) need to be improved to achieve better food security and nature conservation. This is an important example, as land is a finite resource and both meat production and PAs compete for this land."

22. ...instruments, such as...
Done.

23. Would it be possible to expand slightly on this? How asynchronous is the nature of agent decision-making across levels typically?
I am also wondering how this works with the long-term and short-term memory. Perhaps I am not fully understanding something in that paragraph. Would this mean the agent is getting a lot of short-term memory in its prompt and this is saved until the agent acts? Could you comment on that in the paragraph about memory?
We will expand on this yes - please see the response to comment 31 regarding the memory mechanisms of LLM agents.
In the EU governance system, decision-making is often asynchronous, with different institutions operating at different levels. The European Commission, as the supranational governing body, primarily focuses on long-term strategic policy goals (e.g., the European Green Deal, CAP reforms, etc.), setting overarching frameworks for Member States. In contrast, national and regional institutions make more frequent policy adjustments to ensure effective policy implementation and adaptation within their specific contexts.

24. You write the LLM have a polycentric structure. From the text I can't completely visualize this and understand why it is a polycentric. It seems to me, for example, that quite a lot of power still lies with the high-level institution, so I don't see the polycentric structure completely myself when observing figure 1. Could you perhaps describe why you consider the structure to be polycentric?
Although the high-level institution does imply some hierarchical decision-making, the structure represents polycentric governments in the real-world. The operational institutions are relatively autonomous bodies that can make decisions on their own and operate independently, yet interact with each other. The design of the model's polycentric structure was influenced by the real-world example of the European Union, which has also a high-level institution (European Commission), but decisions on implementation are done through autonomous member-state government departments. To improve clarity in the text, we added Box 1 describing the link from the conceptual model (seen in Fig. 1) with the example of the European Union.

25. Could you define this? Does "brain" equal framework?
"Brain" here is used as an analogy to the cognitive architecture of an LLM agent. As you raised this question, it seems this word caused confusion instead of the clarity initially intended. We

have deleted this analogy in the revised manuscript to avoid potential confusion.

26. Could you provide a rationale for using this specific framework? What is the benefit of this one?
We can say a little more, but fundamentally this framework is used because it is useful and sufficient for our purposes, containing extensive elements and being able to represent a range of agent cognitive architectures from simple to complex.

27. Is what you mean by "brain"?
Please see the response to Comment 25.

28. "Synthesizing results" (with a capital for consistency).
Done.

29. Could you define tools? Either in the text or in the caption?
Update: I see you define it later on. Could you, to be able to better read the figure, also define it in the caption of the figure?
Yes, done.

**Page 7:**

30. ...information, such as...
Done.

31. This is the part where my question on the frequency versus memory comes in.
Memory here is not directly related to policy adjustment frequencies. It is more about the technical approaches that determine how an LLM agent handles information. For instance, a prompt with updated information that can be directly input into an LLM constitutes short-term memory. In contrast, information stored in external containers or external files—such as datasets containing historical land use outcomes or a knowledge base with a subset of EU policies—is typically considered long-term memory. Long-term memory reduces reliance on the context window (the maximum number of tokens an LLM can process at once) by enabling the storage and retrieval of vast amounts of information beyond what the model can handle in a single input. We will expand the explanation here and include a citation to a comprehensive survey on the memory mechanisms of LLM agents in the revised manuscript for the reader's reference.

32.    How does the agent evaluate this exactly? I am not sure if I understand well the interaction between, I assume, quantitative output from CRAFTY, that shows whether or not policy goals are being reached etc. and the translated qualitative information the agents receive in which this is reported to them.
For instance, if there is a policy goal on expanding protected areas to 30% of EU land cover, would the agent receive a qualitative "true" or "false" based on a calculation of the extent of protected areas in CRAFTY? For clarity perhaps it is useful to provide an example.
We did not instruct the agents on how to know whether the policy goals were achieved in the experiments. We intended to give the agents high autonomy to analyze and interpret the data in order to let the agents expose their "intelligence" level while keeping the prompts simple. Tables B4 and B5 cited in the experimental setting section have shown the prompts for the agents. As shown in the results, the agents did not do well in this. They compared the average

values of the time series of the land use outcomes with the policy targets, which is not correct. However, this can be avoided by simply prompting it to use the latest data points.

33. In my opinion, the structure of this Section can be improved. I have provided comments that could help guide this process. I general, I suggest to put in one paragraph per agent/actor type and discuss the targets and other parameterization for these agents in that paragraph instead of splitting it up.
Thank you. We will revise the manuscript as per your suggestions.

34. It would be helpful if CRAFTY-EU and the RCP2.6-SSP1 parameterization summarized in the supplementary materials.
We have included the reference to the original publication for information, but can summarize in the supplementary materials.

35. Could you reflect on the potential limitations that may arise from focusing on the EU only? Many social-environmental dynamics in regions outside the EU affect agents and actors as well as policies in the EU. What is the consequence of not taking interactions with regions outside the EU into account? Or is this considered? If it is considered could you please describe this and how this influences the experiment?
Applying this state-of-the-art approach to broader regions will surely be meaningful. But at this exploratory stage, we are not aiming at drawing very region-specific conclusions to guide real-world policymaking. Instead, the focus of this research is on exploring the potential of using AI agents to simulate institutional actors within the existing land use model. We focus on building AI agent workflow, AI agents' capability of autonomous decision-making, exchanging unstructured data, and constructing complex relational structures in institutional networks, as well as how they should be coupled with programmed systems. The evaluation of behaviours is focused on their contextual awareness and logic consistency. We use EU as it provides a nice example for multiple levels of governance, and relates to the spatial coverage of the CRAFTY model used. This research can of course be applied to other areas in future research.

36. Why this scenario type? Why is this a useful scenario to experiment with?
Please see the response to Comment 39.

37. Why simulate until 2076?
This is purely because of data availability.

38. You forgot a dot at the end of this sentence.
Dot added.

39. I do not completely understand the rationale behind the specific experimental settings and parameterization you describe here. Likely this is very understandable of course because you build on previous studies. However, it would help to include a bit more rationale for the choices you make. Why is it useful to consider PAs and meat production? Is it because they have contrasting lobbying groups and operational actors with opposing interests? Why is it useful to consider RCP2.6-SSP1? What is the effect choosing this particular experimental setting over another? I personally think the paper would be a lot stronger if this is better explained and understood by the reader. How can I interpret the outcome of this experiment in context of other types of experiments that could be done and other types of RCP-SSP combinations? Etc.

Thanks for the suggestion. The scenario was chosen really to provide a background to initialize the land use model. The scenario assumes a relatively modest climate change alongside gradually improving socio-economic conditions, including steady economic growth. Under these conditions, the CRAFTY land-use model produces a gradual increase in ecosystem service supply, providing a relatively simple and straightforward baseline for us to investigate the impact of LLM agents. The focus here is more methodological and conceptual rather than empirical. In addition, the agents' believable behaviours should be independent of specific scenarios. The agents are expected to perform reasonable, logically-consistent, and contextually-aware actions to steer the specific model outcomes towards the target level. We choose to incorporate meat production and PA expansion because this provides a conflicting environment. Also, meat production and PA expansion represent two distinct policy instruments – economic incentives and area-based regulations.

We previously used this scenario in our publications to evaluate different approaches to autonomous policymaking agents. While this choice ensures consistency, we acknowledge that exploring alternative scenarios, such as SSP3 and SSP5, would offer rich insights into institutional and land-use dynamics.

We will add clearer rationales throughout.

40. What might work well here to improve clarity and structure is if you split up the pieces per agent type up in the same way as you do in Section 2.1 with italic agent types followed by the experimental setting for that agent, including also immediately the target setting and other parameterization. Of course not necessarily in the same order as Section 2.1 but in an order that works best.
Thanks for the suggestion; we will consider rearrangement for clarity.

41. In the context of this experiment, what would be real-life equivalents of these institutions? Could you give examples? I think it would help to make things less abstract if the reader can keep examples of existing institutions in mind.
An example that illustrates the institutions and governmental structure of the model is the European Union. The operational institutions could be seen as member-state governmental departments. We added Box 1 including a reference to the operational institutions as follows "National scale implementation is done through various government departments that are usually responsible for a specific policy sector, e.g. agriculture, environment, research, etc. Within the model, these government departments are represented by the operational institutions, and the multiple instances of the operational institutions represent the different policy sectors.".

42. ...instruments, such as...
Done.

43. Long sentence, a bit difficult to follow. I suggest to split up or shorten.
Done.

44. Replace "They" with "The operational institutions" for clarity.
Done.

45. You mention the targets later on. I was immediately wondering about the specific targets.

It would help the reader if you move the piece of text about the targets to this paragraph.
Done.

46. Could you provide a real-life example in the context of this experiment?
For example, the European Livestock Voice group advocate for policies that support the sustainability and economic viability of the European meat industry, whilst organizations like EUROPARC seek to secure funding and favourable policies for the conservation and sustainable management of protected areas in Europe.

47. I am a bit confused here whether this piece about the high-level institution applies only to the lobbyist you discuss in this paragraph or whether this is a more general statement that concerns all agents. If the former is the case, could you specify? If the latter is the case, perhaps you can put this information in a seperate paragraph?
Good suggestion. Done.

48. Like for the operational institutions, could you perhaps provide a real-life example of high-level institutions that could play a role in the context of the experiment? This would make everything less abstract.
A real-life example of the high-level institution is the European Commission. We have added a reference to the high-level institution in Box 1 as follows: "The executive body of the EU is the European Commission, which is responsible for enacting new legislation. This is equivalent to the high-level institution in the model. The European Commission proposes new Directives and other policy measures that are ratified by the European Parliament, but which are then implemented by the member states (national governments)."

49. Which agents? Only the lobbyists discussed in this paragraph or more broadly?
We have divided the original paragraph into two to make things clearer.

**Page 8:**

50. Merge with the paragraph on lobbyist.
Additionally, could you describe the experimental settings of the research supplier? (I guess this is considered an "advisor" actor?)
51. Suggestion to put this sentence completely at the end, after you have described the experimental settings of all actors that are involved.
52. Move to part about operational institutions.
Response to Comments 50 - 52: Thank you. We will follow your suggestions in the revised manuscript.

53. You mean the operational actors here right?
Yes. We have modified the text here to improve clarity.

54. What is the rationale behind this? Could you describe this? In real life it is not usually the case, so I am wondering how realistic this setting is.
I can imagine setting an unequal budget allocation could really influence the results and also the reliability of outcomes. Could you reflect on this in the discussion?
We can reflect this in the discussion yes. The main reason is that an unequal initial budget allocation would be harder to justify at any specific level than an equal one under the hypothetical experimental settings while making the results harder to interpret. In future work,

we hope to work alongside policy makers and other stakeholders to parameterize the budget allocation using real-world data, to be able to simulate how funding disparities impact the achievement of policy targets. This is not trivial, as budgets for individual policies are not clearly documented and funding to reach specific targets tends to come from multiple sources.

55. Transfer this statement to a place in the previous text where you talk about budgets.
Good suggestion. Done.

56. I did not know what this is and what is meant by it. Perhaps other readers might also not know it. Could you use different terms to describe this?
Done. Tokens are the basic units encoded to train large language models. Tokens can be derived from words, sub-words, characters, etc. They are normally shorter than complete words.

57. I am guessing "Llama-3-70b-8192" and "gpt-4" refer to things in this library. Could you expand on them here?
Done. They are different versions of models. We have changed the title of the column named "LLM" to "LLM version" to improve clarity.

58. ...behaviour, as well...
Done.

59. A dot is missing at the end of this sentence.
Done. Dot added.

60. It would help perhaps if you put in one extra column in which you specify the agent type (operational, lobbyist, advisor etc.).
Good suggestion. Done.

61. Sometimes you use "Goals" and sometimes "Goal", while it is often not super clear whether what is described is one or multiple goals. I would suggest to be consistent.
Good suggestion. We have changed the word to Goal to improve consistency.

62. I see that indeed quantitative CRAFTY data is transformed into qualitative text. I am still wondering a bit how, for example, things like "we reached the policy goal" is communicated qualitatively. Is this very consistent for each target (e.g., TRUE / FALSE output from the tool? Or is reaching the policy target up for interpretation by the agent based on more vague text?
As mentioned in our response to Comment 32, we did not provide specific instructions on how the agents should determine whether the goals were met. Instead of directly converting data into text, the data were processed using code written by the LLM agents. This code produced numerical results, which were then incorporated into the prompt for the LLM agent to interpret. The agents had full autonomy in deciding how to process and interpret the data, using any relevant computational approach. For example, an agent might calculate the gap between policy goals and actual supply to assess progress. However, the interpretation was not strictly constrained to numerical analysis, as we did not specify in the prompts whether the outputs should include quantitative elements.

63. Why are there different LLMs? Could you describe a bit what is done in each of them and how they differ?

Because Llamma is free but gpt-4o costs money, and both of them have a rate limit per minute imposed by the LLM providers. If we call the LLMs too frequently, the API will report an error, which will disrupt the simulation. So, switching between models can avoid the rate limit and save costs. We will explain this more explicitly in the revised manuscript.

64. Could you define this? What does intermediate output mean and what is done with it?

When an AI agent is solving a problem, it often breaks the task into steps instead of giving an answer right away. The results from these steps are called intermediate outputs. Figure 3 illustrates a sample of these intermediate outputs. We will also explain this concept explicitly in the revised manuscript.

**Page 9:**

65. Perhaps make sure the column width is adjusted to the width of the words.

Sure. We will adjust the column width to better present the words.

66. How is it known what budget is needed? Does this change over time etc.?
Update: I have found information in the Appendix on budgeting. I suggest to move information to the main paper as it is so fundamental to understand the methods and results.

Thanks for noting this. We found that incorporating the budgeting information from the appendix into the main text would make the manuscript unnecessarily lengthy and potentially disrupt the reader's experience. The equations and symbols are interdependent, and relocating the budgeting details would require moving all related technical content as well. To maintain clarity and accessibility, we aim to keep the main text more conceptual while consolidating the technical details in the appendix. This approach ensures that readers with varying levels of interest in the technical aspects can engage with the material at their preferred depth. We appreciate your suggestion and hope this explanation clarifies our rationale. We will refer to the Appendix wherever needed to better guide the readers' attention.

67. For all other institutions (e.g., operational institutions) names were provided, such as agricultural institution. It would be useful if you do the same here for the high-level institution. It would also be helpful if you could put some examples in the text of what a real-life example is of such institution in the context of the experiment.

We have added Box 1 to provide an example of a high-level institution, such as the European Commission. However, directly naming this agent the European Commission could be misleading, as real-world supranational decision-making bodies in the EU context are much more complex than the current agent intends to represent. To maintain flexibility and avoid misinterpretation, we have chosen to keep the institution's name generic throughout the text, allowing future, more empirically focused research to incorporate necessary details.

68. In my opinion, it would be good to describe general results of CRAFTY before going into the other outputs. I find it difficult the interpret the results of the textual outputs without having a context on what was happening with LU in CRAFTY and how this was influenced by the actors. For instance, in the last paragraph of lobbyist it is stated that: "Its discourse primarily focused on two aspects: the environmental threats posed by meat production and the critical importance of conservation efforts. This concern was underscored by the research supplier's data interpretation showing a widening gap between meat demand and supply." I see here that there is a link to the output of CRAFTY but such things are only sporadically mentioned so it

is difficult to interpret the output for the institutional actors in the context of LUC and how they influenced this process in CRAFTY (hopefully this is clear enough to understand what I mean..).

Thanks, we understand your questions. We will highlight CRAFTY results as suggested where essential to understanding the results presented here, and otherwise refer more clearly to previous publications where extra details can be found.

69. Please define what this means. I would suggest you discuss in detail how you test for this in the methodology. Or highlight what criteria you use to qualitatively interpret the behaviors etc.

Erroneous behaviours means any unintended or incorrect actions performed by an LLM agent. In the manuscript, erroneous behaviour includes misinterpretations of data, incorrect formatting, logical inconsistencies, or failures in workflow execution. We examined all the agents' output manually, intending to capture mistakes or errors generated by the agents. There is no standardized or established method to evaluate agents' outputs especially when their outputs are unstructured and dependent on the logical validity. In the Results and Discussion sections, such behaviours have been highlighted. Some mistakes are straightforward. For instance, the research supplier agent failed to generate a final output because its "thinking" process used too many iterations or too much time; The high-level institution agent occasionally failed to follow the instructions to consider all other agents' input. Some mistakes are less obvious. For example, the operational institution agents were found using the mean values of the times series rather than the latest model outcome to evaluate how close the current outcome is to the policy goals. This mistake requires us to examine the intermediate outputs carefully to see how the data were used.

70. Please define (see previous comment)

We will amend and explain more explicitly.

71. Here you split up nicely between the institutional types. I would suggest to also do this in the experimental setting section. Perhaps even in this exact order.

We will reorganize the text in the experimental setting section while avoiding repeating the information in Table 1.

72. This advisor is not so much described in experimental settings. Would be useful to have a bit more background.

Table 1 has highlighted the input, action, output, and goal of the research supplier agent. The agent's prompt (Table B1) provided well-formatted information to define what the agent's role is. We will cite Table B1 in Table 1 to better guide readers' attention.

**Page 10:**

73. This seems to me something that belongs in the methodology. It is not described there. I would suggest to describe it in the methodology and only repeat it shortly here.

Done.

74 ....mitigation".

Done.

75. To me it was not clear that other laws and policies that those directly regarding PAs and meat could influence things. This makes the experiment a bit broader that I expected. Could you state this explicitly in the methodology? To which extent do other policies not directly related to PAs and meat have influence? I see here the agent is interpreting a nature restoration policy in the context of PAs even though it does not directly apply to PAs. In this case it makes sense to a certain extent, even though I am not sure how this would influence PA expansion efforts eventually (would it shift to protecting degraded areas with the aim to restore them and would this then reduce the area of, for instance, intact forest to be restored?).

Thank you. This is an interesting point. It is true that the influence of policies beyond those directly targeting PAs and meat production has not been explicitly stated. In the model, institutional agents, particularly the law consultant and high-level institution, can interpret a range of policies and regulations in ways that extend beyond their immediate scope. This reflects real-world policy-making, where broad legislative frameworks (e.g., biodiversity strategies, restoration policies, and climate regulations) can indirectly shape sector-specific decisions.

Indirect policies can shape the agents' narratives and lead them to propose actions beyond what is explicitly represented in the land-use model. However, the agents have limited direct influence on the land-use model's operations: the high-level institution can only adjust budget allocations and policy goals, while operational institutions can only impact meat supply or PA coverage. As a result, policy actions suggested in the agents' textual outputs do not translate into direct interventions that alter the land-use model's behaviour. A clear example, as noted in the manuscript, is the high-level institution's proposal to allocate 30% of the total budget to "other initiatives and programs", a policy action not represented in the model.

Assessing how textual variations in inputs affect the numerical results of the land-use model is inherently challenging, given the vast number of possible word combinations. As a general guideline, it is advisable to select only the most relevant policies for agents to reference, as excessive text may dilute the LLM's attention and hinder reasoning performance (Levy et al., 2024).

We will highlight this point in the discussion.

**Page 11:**

76. This is quite logical right? This is inherent to the experimental setting described in Table 1. Or does the high-level institution have the power to create new "EU laws, policies, regulations, etc."?

Correct, this agent is not able to create new laws. It can only select a subset of policies in the static knowledge base. The law consultant agent behaved as expected because this agent did not receive enough updated information. The law consultant agent is designed to offer a set of constraints that may modify the high-level institutions' behaviour. A more sophisticated law consultant agent could be built in future studies with more dynamism.

77. Sentence seems incomplete. ... in other years seems to be missing. I suggest rephrasing.

We think this sentence is adequate as it is, but will check for clarity in the context of the revised version.

78. It would help the reader if this figure is positioned after the paragraph or the piece of text that refers to this figure (last paragraph of Section 3.1.2).

Done.

79. This is not explained in the methodology. I see you explain it in the results below. I suggest to explain such things in the methodology.
We will try to follow your suggestion and add a description of these analysis methods to the methodology section without disrupting the text flow.

**Page 12**

80. Much of this belongs in the methodology.
We will reorganize the text accordingly.

81. In the above result section often such statements are made but I don't understand so well where they come from and how to interpret them in the context of what is described in the methods and in the context of the experiment. As far as I understood, the input to the lobbyist agents is the research supplier's output and the profile. In that case, is it specified in the profile that the lobbyist always advocates for increased budgets? It is described that the agents compete for budget but not how the budget is used to implement policies (how expensive are things, how do agents know that they don't have enough budget, or do they always want more budget anyway?)
There are more such statements all throughout the result section. In my opinion, it would be helpful to go through the statements and see if all information needed to interpret them is provided in the method section.
Please see the response to Comment 82 below.

82. I see here also most output concerns budget challenges. Could you describe this a bit more in the experimental settings? How do budget challenges arise? And how is it determined how much it costs to implement certain policies etc. I expect this data comes from CRAFTY?
In general, it would be good to describe budget allocation and use. How, for example, can a budget be inefficiently allocated within the context of the model?
Response to Comments 81 and 82:
Tables B1 and B2 show the prompt templates of the lobbyists. They are not instructed to advocate for increased budgets but to prompt the high-level institution to prioritize nature conservation or meat industry development.

Regarding the budget challenges, the prompts in Tables B4 and B5 instruct operational institutions to seek increased priorities and financial support for meat production or protected area establishment, and to prompt the high-level institution to set policy goals that align with budget allocations. Consequently, agents are motivated to request additional budget if policy goals are unmet.

The costs of policy adjustments are determined via Equations C2–C7, which evaluate policy performance and the resources required for subsequent adjustments. Table C1 provides the parameter settings for policy-resource functions, indicating the budget needed for corresponding policy adjustments. When one institution consistently experiences a surplus while another faces a deficit, it signals inefficiency. This can be addressed by reallocating funds or adjusting policy goals to better align with available resources, improving overall financial efficiency and policy effectiveness.

We will explicitly explain and cite the information from appendices in the main text to better guide readers.

**Page 13:**

83. Here again policies come into play that do not relate directly to PAs. It would be good to understand how the agents combine the policies that directly apply to PAs to those indirectly applying to PAs. In principle, net-zero targets are not direclty related to PA expansion in EU, protection of biodiversity is.
Please see the responses to Comments 75 and 104. We will discuss this in the revised manuscript.

84. Is this what is meant by "potentially erroneous agent behaviours"?
Yes, this is one of the erroneous agent behaviours.

85. What about the researcher agent? Did this agent make the same mistake? Did this lead to inconsistent supply of input to the high-level institution?
Good point. Unlike the operational institution agents' outputs, there is no numerical information explicitly mentioned in the agent's interpretation of data analysis. This might be because these agents used different large language models with different prompts, which could influence the agents' "intelligence". However, it is reasonable to suspect that the researcher agent could also make the same mistake. A safe way to address this issue is to explicitly instruct these agents to use the latest data points to determine the current state of the land use outcomes.

86. You mean here that the agent did not change their conclusions based on this error?
Yes. Gaps exist between the policy goals and actual outcomes (meat supply and PA coverage) in most years. Thus, the mean values of the actual outcomes are lower than the policy goals, which is qualitatively consistent with comparison results using the latest data points. This mistake needs careful examination of the intermediate outputs. Otherwise, it would be more evident to capture and easier to correct during prompt design.

87. involved
Corrected.

88. Could this be related to the short-term vs long-term memory?
This could be caused by the failure of the LLM to follow a very lengthy prompt. Please refer to the response to Comment 31.

89. I see here the policy actions and outcomes are described. It would be quite informative to put these results prior to the ones on the institutions output to provide background necessary to interpret the output.
Or perhaps the PA changes, LU changes and meat production changes can be discussed prior to the institution output, and the feedback with the institutions and the CRAFTY output either in the institutions output section or a separate section below.
I feel though quite some pieces in this section arguably belong in the discussion. I have highlighted them.
Thanks for the suggestions; we will revise for flow and clarity.

90. Which assumptions were made for PA allocation? Would be good to specify.

We have cited the corresponding text in appendices to better guide the reader's attention.

91. This seems to belong in the discussion.
We will consider moving this to the discussion or removing it.

92. This paragraph contains a combination of some results with a lot of discussion. I suggest to move it to the discussion. And expand more broadly on the results for PAs, meat production and LU prior to the institution outcomes.
While we have limited room to expand all the points raised in the reviews, we will try to improve this section including by moving text to the discussion.

93. Repetition.
We will check both of these points for repetition.

94. Repetition
As above.

**Page 14:**

95. Could you define "Gap" in the figure caption?
Done. The gap means the difference between the policy goal and the actual outcome (e.g., meat supply and PA coverage).

**Page 15:**

96. Maybe rephrase as: ... resulting in a sum of the budget allocation ration of 0.7.
Good suggestion. Done.

97. This seems more fitting for the discussion.
The textual output and numerical analysis are matched as the latter is the outcome of the former. We will rephrase the text here.

98. In order for the reader to judge whether behavior of LLM agents is believable, "believable behavior" needs to be described and defined. I suggest to do this in the methodology (see earlier comments). Additionally, there seems to be a mix of terms: believable behavior, erroneous behavior, counter-intuitive behavior etc. It may be useful to be consistent. As none of them are defined in the text the readers are left to interpret them for themselves, in my case resulting in different interpretations and confusion.
We will clarify their meanings.

Believable behaviour means that an agent's actions and decision-making processes resemble realistic human behaviour.

Erroneous behaviour indicates unintended or incorrect actions performed by an LLM agent. In the manuscript, erroneous behaviour includes misinterpretations of data, incorrect formatting, logical inconsistencies, or failures in workflow execution.

Counter-intuitive behaviour refers to an emergent pattern or decision-making outcome that deviates from conventional expectations or common sense. In the context of the manuscript, counter-intuitive behaviour occurs when the high-level institution's decisions do not align with typical policy-making norms, such as the unexpected budget allocation, which may contradict the assumption that the percentage of the total budget should be 100%. This behaviour does not necessarily indicate an error but rather an unanticipated outcome driven by the agents' decision-making.

**Page 16:**

99. ...tasks, such as...
Done.

100. ...challenges, such as...
Done.

101. Ok now I understand. Please describe this in the methodology. How is the high-level agent making these decisions? How does it know how expensive things are etc.?
This has been described in the methodology (please see Table 1 and Table B7). The decision-making details are presented in the equations in Appendix C. We will cite them explicitly in the methodology section.

102. This sentence does not flow very well. I suggest rephrasing it.
Done.

103. Perhaps I don't have enough knowledge on EU dynamics, but this outcome does not seem very intuitive to me. In general there is not enough funding to protect nature worldwide (I am not an expert on EU though, so perhaps I am very wrong in this context).
Would it be possible to discuss the outcome in context of past dynamics and prior modeling studies? To discuss whether it is realistic that the environmental institutions had so much influence?
It seems almost like the opposite of real life is happening, where policies are very influential (even though that is often not the case, for instance in EU already around 1 billion trees out of the 3 billion should have been planted but we are at 22 million) and the wishes of stakeholders on the economic and livelihood side of the spectrum have almost no power. In the case of The Netherlands, the opposite occurs currently, where the state has taken very limited action on, for instance reducing nitrogen, not reaching the target by far.
Perhaps you can discuss the findings in the context of such examples (but more fitting ones)?
Please see the response to Comment 105 below.

104. Here the other indirect policies come into play. Could you further discuss on the interactions between direct and indirect policies related to PAs and meat production? Do the agents understand the difference?

Thank you for your question. Understanding the interactions between direct and indirect policies related to PAs and meat production is a highly complex task. Analyzing such interactions in depth would require a dedicated study, as it involves effort beyond the current scope of our research. While our model captures how institutional agents process and interpret

various policies, it does not explicitly distinguish the causal relationships between direct and indirect policy impacts. Future work could explore this issue more systematically by designing experiments that isolate specific policy interactions and their cascading effects on land use changes.

The extent to which LLM-powered agents "understand" the distinction between direct and indirect policies requires careful consideration. Our approach does not assume that LLMs possess genuine comprehension but rather utilize their ability to process textual patterns and generate human-like reasoning. Below, we clarify how our agents operate in relation to distinguishing policies:

*LLMs and policymakers can distinguish textual patterns.* LLMs are pre-trained on vast amounts of text and good at recognizing textual patterns. If the policy instruments and targets have strong textual relevance, LLMs should be able to differentiate between direct and indirect policies, just as human policymakers do. Policymakers rely on experience, legal frameworks, and historical precedents, while LLMs rely on learned associations from their training data.

*Textual relevance alone is insufficient for effective policymaking.* However, effective policymaking is not just about distinguishing textual relevance, It is about predicting and evaluating policy outcomes. Understanding the real-world impact of policies, including their unintended consequences, is a highly challenging task in complex socio-ecological systems. Given these complexities, we did not expect current LLMs to outperform human policymakers in making policy impact assessments.

*LLMs do not truly comprehend, but they mimic human reasoning.* While some researchers have suggested that sophisticated reasoning capabilities in LLMs could emerge through reinforcement learning (often referred to as the "aha moment" (Guo et al., 2025)) current evidence indicates that LLMs still lack true comprehension of policies and their effects. Instead, they utilize contextually meaningful natural language to mimic human reasoning. While this approach allows them to generate plausible and logically structured arguments, it is not free from hallucination (factually incorrect or fabricated outputs). That said, LLM technology is evolving rapidly, and the rate of hallucination has significantly decreased, which also suggests that the logical coherence within their outputs has improved over time (please see the Hallucination Leaderboard).

*Fine-tuning or prompt engineering could improve distinction, but we chose autonomous decision-making.* LLMs can be fine-tuned or prompted by end-users to better distinguish direct and indirect policies, improving their reliability in specific policy contexts. However, in this study, we did not manually engineer prompts to enforce such distinctions. Instead, we allowed the LLM agents to autonomously decide how to interpret policies, as the goal of our research was not to design highly optimized LLM agents but rather to evaluate both their potential usefulness and their limitations in institutional decision-making.

We will discuss these points in the revised manuscript.

105. In real life you often see exactly the opposite. It seems to me that environmental policy documents have too much power and are too influential in decision making about budget allocation. Could you reflect more on this using prior studies on European context?
The numerical results show that the environmental institution in our simulations was over-funded, while the agricultural institutions were struggling with an inadequate budget. This

diverges from intuitive expectations based on real-world dynamics and is caused by the parameterization assumed (see Table C1 for details): the budget settings are hypothetical and are roughly set to make the budget needed for different policies the same scale and comparable. Future efforts could focus on calibrating the model with empirically accurate data as the research evolves toward empirical analysis and the development of actionable policy recommendations.

Globally and within the EU, environmental policies and initiatives often face significant funding constraints. For instance, the EU's ambition to plant 3 billion trees by 2030 has seen limited progress, with only 24 million planted thus far (*3 Billion Trees initiative*). Similarly, challenges in reducing nitrogen emissions in countries like the Netherlands highlight the limited influence of environmental policies in the face of economic and political barriers.

This imbalance was also prompted by the environmental institution and the research supplier misleadingly informing the high-level institution that PA coverage was positively correlated with budget surplus. Indeed, both the two operational institutions insisted that their respective policy targets (PA coverage and meat production) should be increased because those targets were positively correlated with other desired outcomes. While these mechanisms aim to capture real-world lobbying behaviours, they may overestimate the environmental institution's effectiveness in securing resources.

The influence of environmental concerns might also come from the biases in the LLMs' training data, as text containing social norms favouring environmental protection over economy might play an important role in the training process. LLM biases have been well documented in the literature (see e.g., Zhou et al., (2024)) and can be rectified by prompt design and fine-tuning (Taubenfeld et al., 2024; Tao et al., 2024).

To align the findings with real-world contexts, future iterations of the model could incorporate a more nuanced representation of political and economic pressures. This would include explicitly modelling the comparative lobbying power of agricultural stakeholders and the structural barriers faced by environmental policies and organizations, in part through unequal budget distribution. By doing so, the model could offer a more balanced reflection of institutional dynamics, providing deeper insights into the challenges of achieving sustainable policy outcomes.

We will reflect on these points in the discussion section.

106. Could you please expand on this? I think this is fundamental and highlights one of the "counter-intuitive or potentially erroneous agent behaviours". Could you for example reflect in more detail on how an over-reliance on policy documents to inform decision-making by the high-level agent could be prevented?
Yes, we will discuss LLM biases in the revision.

107. I don't fully understand why you conclude that there is a lack of decisive action by the high-level institution. The budget allocation significantly changed and there was a lot of change in PAs in response to the actions of the high-level institutions (at least this appears to be the case from what was written in the results). Could you reflect on this and indicate what you mean with a lack of decisive action in this context?
It is true that there were significant changes in PAs over time. However, these changes were

primarily driven by the availability of surplus funds rather than strategic intervention by the high-level institution. The environmental institution had no budgetary pressure, allowing it to continuously expand PAs. In contrast, the agricultural institution consistently experienced a budget deficit, which should have prompted the high-level institution to reallocate more funds towards meat production to balance resource use.

Despite these circumstances, the high-level institution did not make sufficiently large budget reallocations to correct the imbalance. For most of the simulation (40 years), the budget distribution remained 60% for PAs vs. 40% for meat supply, and only in the final 10 years did the ratio shift slightly in favour of meat production (55% vs. 45%). However, this modest adjustment was insufficient to reverse the long-standing budget surplus for PAs and the deficit for meat supply. A truly decisive action in this context would have been a more prominent reallocation of funds—such as shifting the majority of the budget toward the underfunded agricultural sector at an earlier stage.

Thus, when we refer to a lack of decisive action, we mean that the high-level institution failed to make bold, corrective adjustments to its budget allocation, instead following an incremental approach. This behaviour aligns with path-dependent decision-making, where institutions adjust policies gradually rather than making radical shifts, even when inefficiency persists.

108. I am personally not very convinced that the behavior is believable (also here the term "believable" is rather abstract and not defined). A big budget surplus for environmental issues compared to agriculture seems quite counter-intuitive to me. As far as I am aware, the opposite is usually the case.
Could you discuss this please in context of past and current dynamics in the EU and prior studies?
Please see the response to Comment 105 for budget surplus issue.
The believability of agent behaviours in our model is determined by whether they capture aspects of real-world decision-making within the given experimental context. The experimental results indicate that the high-level institution was unable to sufficiently reallocate funds from the over-funded environmental institution to the under-funded agricultural institution. This occurred because the institution had to balance multiple, conflicting stakeholder inputs, leading to incremental rather than decisive policy shifts. Instead of effectively correcting the budget imbalance, the high-level institution followed a decision-making process constrained by the necessity of considering multiple institutional perspectives. This behaviour reflects some aspects of real-world policy-making in democratic systems, where competing interests, bureaucratic inertia, and consensus-driven decision-making often limit drastic policy changes (Lindblom, 1959; Jones, 2003). The model thus reflects bounded rationality, where policymakers operate within cognitive and informational constraints, and incrementalism, where policies evolve gradually rather than through radical shifts. While the model may have overrepresented the shift toward environmental funding, it captures the challenges of redistributing resources in multi-actor governance settings, such as those found in the EU's Common Agricultural Policy (CAP), where competing priorities frequently lead to compromise-based rather than optimal budget allocations (Daugbjerg & Feindt, 2017).

**Page 17:**

109. Please define.

Done.

110. What does this mean? Please define.
Done.

111. Reference is missing.
Reference added.

112. Could you perhaps reflect on whether this is realistic? I can imagine this happens quite a lot in real life? Is there an example of this happening in the EU?
We will try to add an example.

113. large-scale land use models
Done.

114. How does this work in case of the high-level institution which operates at lower frequency? Is more often the maximum number of tokens reached? If so, how does this influence the results?
The frequency here does not have direct connections with token numbers. The frequency is introduced to simulate the time lags of policy adjustments, as policymakers in the real world normally do not respond to land use changes immediately (Watts et al., 2020).
The maximum number of tokens that can be processed by an LLM like GPT-4o used in this paper can reach 128000 (see the documentation of the models here), approximately 96000 words estimated based on the rule of thumb that 100 tokens equals 75 English words. Technically, the context window is generous. However, research shows that the reasoning performance of LLMs drops notably as the length of prompts increases even if the length is far less than the technical maximum (Levy et al., 2024), which means the best practice is to use concise prompts; otherwise, the LLM outputs might be more prone to mistakes.

115. It seems useful to put this in the methodology. This is not described there it seems.
We will add to the methods.

116. Perhaps this should be put in the methodology and the bounded rationality argument used as a rationale for the limited size of the context window?
We think this works best in the discussion because it relies partly on results generated here, and so would be less clear earlier in the manuscript.

117. Reference missing.
Added.

118. I am not sure if I would personally consider these errors. I interpret them as inherent aspects of the model and assumptions made, parameterization etc. Perhaps you can call them limitations?
Yes, we can change this.

119. This is useful information for the methodology. It is not described there as such.
We will check that and the point below is covered in the methodology.

**Page 18:**

120. This is all methods that is not included in the methodology. It would be very helpful to

put it there, it is answering some of the questions that came up. When you put it in the methodology, please expand on it. For instance, what is the human-in-the-loop exactly?

Thanks. We will expand on this in the revised manuscript.

121. How exactly is this quantitative data translated into qualitative data by the agents?

Please see the responses to Comments 32 and 62.

122. Would it be possible to discuss the limitations of the modelling approach in terms of the limited number of policy targets? And in general the experimental set up and whether different dynamics between environmental and agricultural agents and lobbyists etc. would be expected when they are fed different policy targets (e.g., 1 billion tree planting instead of PAs). Do you think the same issues would arise?

In other words, following this assessment, is the model sufficiently tested to run other types of scenarios with other policy targets, other institutional agents etc.? Or are more tests and sensitivity analyses necessary?

We can briefly touch on this, but as initial proof-of-concept research, the modelling presented here is not really intended to establish all of these outcomes. Such work would require a range of specific inputs as well as targeted tests. We will discuss the generalisability of our findings, however.

123. I don't fully understand how the robustness is of the modeling approach is tested. There is no sensitivity analysis with different starting conditions and no comparison between scenarios etc. So I personally don't think you can speak of model robustness in the sense as it is a term usually implying some sensitivity analyses etc.. Additionally, the terms "error-proneness" and "error-tolerance" both imply some sort of quantification of the error by the model. In principle it seems that the dynamics that are called "errors" are inherent to the model structure and assumptions and the way agents retrieve information from text. Perhaps you can just refer to them as limitations or "unrealistic dynamics" or something like that?

Good suggestion, thanks - we think this will indeed help to clarify the work. We'll check terminology throughout with these points in mind as well.

We'd like to clarify that robustness here refers to the ability of the institutional network model to function effectively despite errors, incomplete information, or unexpected disruptions. It is understood from the perspective of system science. A robust system maintains functionality even when agents produce erroneous outputs or misinterpret data. This type of robustness does not imply statistical robustness (which requires sensitivity analysis) but rather operational resilience, meaning that the model does not collapse or produce entirely unrealistic behaviours due to minor errors.

124. Didn't the environmental agents underestimate PA extents and wouldn't this have lead potentially to more power to influence budget allocation to increasing protected areas?

Underestimating PA coverage alone does not inherently grant environmental agents more power. Since actual PA coverage is also underestimated, the gap between policy goals and reality may not necessarily be larger. The influence of environmental agents depends on multiple factors, including the narratives they present, the counterarguments from competing agents, and how the high-level institution weighs inputs from other LLM agents.

125. I already wrote a comment on this before. It does not seem to me that the high-level institution is making balanced decisions, particularly not on the budget division part. Perhaps though I don't know what you mean by balanced in this context. Perhaps you could define what

balanced means to you?

In this context, "balanced decisions" do not imply the final budget surplus/deficit accumulated in different operational institutions. Instead, it means that the high-level institution avoided extreme reallocations because of the conflicting inputs from different stakeholders, which reflects incrementalism and path-dependent decision-making, common in real-world governance (Lindblom, 1959; Jones, 2003).

We have rephrased the sentence in the revised manuscript to improve clarity: The high-level institution's tendency to seek compromise among competing policy priorities contributed to the error-tolerance of the institutional network.

126. I do not fully understand how you derive this conclusion about distrust in simulations.
This does not refer to distrust in simulations, but between policy actors; we are simply drawing a parallel to a real-world issue here.

127. Perhaps replace the arrow with writing?
Done.

128. This is why a definition of error is needed. In my opinion the LLM output is not correct nor incorrect (unless there is a bug or something fundamentally wrong with the simulations). The model simulates into the future, so it can never be correct in that general sense. Additionally, if "correct" means that agents never make mistakes, this would not be realistic.
In the context of the discussion here, "correct" means logically correct. It means the resultant policy decisions or actions are derived logically from the reasoning process conducted by the agents. In the revised manuscript, we have made this explicit.

**Page 25:**

129. There is quite some information in this section that could help clarify things for the reader in the main text. If possible, I would suggest to transfer some of this information so that that reader can more easily follow the methodology and interpret the results.
Please see the response to Comment 130.

**Page 26:**

130. Could you transfer this to the main text? I was missing information like this to interpret the results.
Thank you for noting this. This is very difficult to do without creating further confusion, as much of the text depends on other sections for meaning. We will cite the equations here explicitly in the revised manuscript for clearer guidance as per your suggestion in *Summary of the review*.

**Page 27:**

131. Perhaps it is possible to refer a bit more explicitly to these sections of the SI by stating for instance that you can find information about the policy implementation for PAs in this part of

the SI in the methodology. This would be very helpful because then the reader would know that the information is provided somewhere. Now this is not included in the experimental settings.

Good suggestion. Done.

**Page 31:**

132. References
Done.
* * *
**Box 1.** *How the stylised model and experiments map onto real-world policy institutions in the context of the European Union.*

The stylised model and experimental design presented here were inspired by the real-world mechanisms for policy delivery within the European Union (EU). Whilst the EU reflects a specific set of policy institutions and policy instruments, many of these concepts are transferable to other parts of the world with similar governance modes. In this box, we outline the relationships between the model components, especially the modelled agents, and their real-world counter-parts that are outlined in Fig. 1.

The executive body of the EU is the European Commission, which is responsible for enacting new legislation. This is equivalent to the *high-level institution* in the model. The European Commission proposes new Directives and other policy measures that are ratified by the European Parliament, but which are then implemented by the member states (national governments). National scale implementation is done through various government departments that are usually responsible for a specific policy sector, e.g. agriculture, environment, research, etc. Within the model, these government departments are represented by the *operational institutions*, and the multiple instances of the *operational institutions* represent the different policy sectors.

Beyond the basic mechanism for policy implementation described above, policy institutions are influenced by a number of external bodies. Within the model, these are the *lobbyists* and the *advisors*. In the European context, lobbyists could include land owner associations with responsibility for the economic well-being of their membership. They also include environmental Non-Governmental Organisations (NGOs) that lobby for stronger environmental protection. *Advisors* can include lawyers who support the legal aspects of policy development and implementation as well as scientific researchers who provide policy institutions with knowledge to support policy development (at least in principle). It should be noted that the European Commission, a *high-level institution,* is also a very large research funder, providing financial support for policy-relevant research in universities and other research organisations across the EU.

The stylized experiments applied to the model, also reflect a real-world context. Protected Areas (PAs) currently cover 26% of the EU's terrestrial land surface, and so represent a major land use by area. Likewise, the livestock sector in the EU uses large areas of pasture and cropland to provide livestock feed for meat and milk production. Consequently, livestock farming and nature conservation compete for finite land resources, and policy interventions are a major contribution to resolving this competition. Competition processes are central to land system models such as CRAFTY.

| Terms and definitions within the context of this research | |
|---|---|
| Term | Definition |
| Institution | An organization or governing body involved in policy-making, such as government agencies, research institutions, or NGOs. |
| Institutional Dynamics | The interactions, adaptations, and power relations among these institutions over time, which influence how policies are formulated, negotiated, and implemented. |
| Institutional Network | A structured system of interconnected institutions that interact within the policy-making landscape. |

| | |
|---|---|
| Agent | A computational entity within the model that represents an institution, stakeholder, or decision-making body. Agents in institutional network are powered by LLMs and autonomously make decisions, process information, and interact with other agents. Agents in the CRAFTY land use model represent various types of land users. |
| Actor | A general term for entities (individuals or organizations) involved in decision-making processes. Actors can be both real-world stakeholders (e.g., policy-makers, lobbyists) and their simulated counterparts (LLM-powered agents). |
| Long-term/Short-term Memory | Memory components of an LLM-powered agent that influence its decision-making process. Short-term memory refers to immediately available contextual information embedded in the agent's prompt. Long-term memory is stored information retrieved when needed, using data retrieval techniques such as RAG, allowing agents to reference past information and knowledge base. |
| Tools | External functions or programming scripts that LLM agents can execute to complete tasks beyond text generation. Here, tools include Python scripts for data analysis, information retrieval, and numerical computations. |
| Tokens | The fundamental units of text processed by an LLM, representing words, characters, or sub-words. Token usage is a consideration in computational cost and efficiency when running LLM-powered agents. LLM providers normally charge API users by calculating input/output tokens. |
| Intermediate Output | Partial results generated by an LLM agent before reaching a final decision. The intermediate output allows agents to iterate on their reasoning, refine calculations, and update their responses before producing a definitive action. |
| Erroneous Behaviour | Any unintended or incorrect actions performed by an LLM agent. In the manuscript, erroneous behaviour includes misinterpretations of data, incorrect formatting, logical inconsistencies, or failures in workflow execution. |
| Gap | The difference between policy targets and actual outcomes. For example, in the manuscript, gaps exist between projected meat production levels and actual supply, or between budget allocations and their intended impact. |
| Believable Behaviour | The extent to which an agent's actions and decision-making processes resemble realistic human behaviour. Believable behaviour in the manuscript refers to whether the LLM agents mimic the constraints, trade-offs, and reasoning expected of real-world institutional actors. |
| Hallucination | The generation of plausible-sounding but factually incorrect or logically flawed responses by an LLM agent. In the manuscript, hallucinations occur when agents produce misleading data interpretations, misrepresent numerical values, or invent non-existent policies and regulations. |
| Robustness | Robustness refers to the ability of the institutional network model to function effectively despite errors, incomplete information, or unexpected disruptions. A robust system maintains functionality even when agents produce erroneous outputs or misinterpret data. This type of robustness does not imply statistical robustness (which requires sensitivity analysis) but rather operational resilience, meaning that the model does not collapse or produce entirely unrealistic behaviours due to minor errors. |
| Counter-intuitive behaviour | Counter-intuitive behaviour refers to an emergent pattern or decision-making outcome that deviates from conventional expectations or common sense. In the context of the manuscript, counter-intuitive behaviour occurs when the high-level institution's decisions do not align with typical policy-making norms, such as the unexpected budget allocation, which may contradict the assumption that the percentage of the total budget should be 100%. This behaviour does not necessarily indicate an error but rather an unanticipated outcome driven by the agents' decision-making. |

[revised manuscript text omitted]

---

## Author Comment (AC2)

**Reply on RC2**

- **Black text** indicates reviewer comments.
- **Blue text** indicates author responses.

In the manuscript (submitted to GMD) "InsNet-CRAFTY v1.0: Integrating institutional network dynamics powered by large language models with land use change simulation", Yongchao Zeng, together with his collaborators, has developed a very interesting and powerful technique to use multiple institutional agents each with its own large language model (LLM) prompt history, together with a land-use change model (the latter based upon the CRAFTY model). For the entire European region, they can simulate the inter-institutional dynamics, with unstructured text (i.e., bullet-point recommendations) and numerical output being passed from one institutional agent to another, driving both the changing meat production and the changing percent of land that is a protected area. The agents that are defined are as diverse as a lawyer agent that is familiar with European law, to lobbyist agents that take the side either of agriculture or of environmental advocacy, and further to a high-level institution agent that has long-range goals in mind and that integrates the advice of other agents and prompts the other agents to try to achieve its goals. I am particularly impressed with this paper, never having imagined LLM chatbots that talk to each other, and furthermore never having imagined that these LLM chatbots can be defined with the prompt engineering to groom them as specialist institutional chatbots that can drive a land-use simulator. The writing (grammar, structure, etc.) is of very high quality. I only ask for minor revisions, which I enumerate below.

Dear Reviewer,

We are delighted that you find our work interesting and valuable. Your recognition of the novelty of our work is greatly appreciated.

We also appreciate your constructive comments. We have addressed each of them in the responses below. We believe these refinements will further enhance the clarity of our study.

Sincerely,

Yongchao Zeng (On behalf of all authors)
* * *
Lines 65-66: Holzhauer et al. (2019): This reference is missing in the list of references.

Thank you for pointing out this. We have added the reference in the revised manuscript.

Line 251: Why not SSP3 or SSP5 for the changing climate? SSP1 has little change climate-wise from the current time.

The SSP1 scenario assumes a relatively modest climate change alongside gradually improving socio-economic conditions, including steady economic growth. Under these conditions, the

CRAFTY land-use model produces a gradual increase in ecosystem service supply, providing a straightforward baseline for us to investigate the impact of LLM agents.

We previously used this scenario in our publications to evaluate different approaches to autonomous policymaking agents. While this choice ensures consistency, we acknowledge that exploring alternative scenarios, such as SSP3 and SSP5, would offer rich insights into institutional and land-use dynamics.

We will address this point in the discussion section.

Line 286: how long is an iteration in days or months or years?

Here one iteration indicates a year. We will mention this explicitly in the revised manuscript.

Line 296: What are the differences between the definitions and between performance of Llama-3-70b-8192 and gpt-4o, listed below in Table 1?

Both LLMs used in our experiments were among the most advanced models available for handling textual data at the time. A key distinction is that Llama-3-70b-8192 (developed by Meta) is open-source, while GPT-4o (developed by OpenAI) is closed-source. Both of the models were accessed via APIs that require internet connection. We chose to use both models instead of one due to rate limits on token usage and API calls per minute, which could disrupt the simulation. By distributing the API requests across both models, we mitigated the risk of errors caused by exceeding rate limits. Additionally, this approach helped reduce token costs, as Llama was accessed via the Groq platform, which offers free usage within a certain range.

We will expand on this point in the revised manuscript for further clarity.

Line 301: Table 1: Maybe "Wiring" needs to be defined?

Thanks for noting this. This is a typo. We have corrected it to "Writing" in the revised manuscript.

Line 306: Is this amount of output for the whole period of 2016-2076? Or is it per iteration? I'm a bit surprised that the amount of output is so small. If you're simulating land use over all of Europe with a 5-arcminute spatial resolution, I would expect a lot more output, especially if different countries have different policies.

Thank you for your question. The CRAFTY land-use model is driven by numerous rule-based agents representing different types of land users and produces a large amount of data per iteration. However, the output mentioned in Line 306 refers only to the text generated by LLM agents between 2016 and 2076. Since the LLM agents were activated every ten iterations, text generation occurred only six times during the entire simulation period, with each activation producing an average of approximately 3,300 words.

In this study, we did not model country-specific policies for each EU nation. Our primary objective was to assess the feasibility and proof-of-concept of integrating LLM agents with the rule-based land-use model, focusing on their contextual awareness and logical coherence, which requires

manual examination of textual outputs. Incorporating individual country policies would significantly increase data volume, making manual analysis impractical.

We acknowledge the importance of country-specific institutional agents and the challenges associated with scaling up to integrate them effectively. To address this, we are collaborating with experts in political science, and this will be a key focus of future research.

We will reflect on this point in the discussion section to provide further clarity.

Line 371: What does a "link" between nodes signify in a word graph?

In a word graph, a link between two nodes indicates the two connected words appear within a specified proximity to each other in the text. Proximity can be understood as a window that identifies two words represented by a pair of connected nodes in the graph. The window size was set to 4 words in our analysis, indicating we connected the words within the proximity of four words. The thickness of a link means the frequency of two words coexisting in the same window across the whole text.

We will expand on this briefly in the revised manuscript.

Line 500: If you don't discuss this elsewhere here in this paper, it might be useful to know: how much your computers or the LLM computers need to work to produce these results? And how long from start to finish does a simulation take?

Thank you for your interest in the technical details. We did not record the exact runtime, but a full simulation could take several hours on a laptop with 32 GB RAM and 12 CPU cores (Intel i7-1260P). This is because we used an emulator to run the land-use model, which simplifies the computation required by the original software. One challenge was that the LLM-generated output did not always conform to a valid JSON format, occasionally requiring manual corrections before the hard-coded land-use model could process the data.

As LLM technologies have rapidly evolved, output reliability has significantly improved since our experiments. However, applying LLMs to simulate a large number of agent interactions (such as those involving over 20,000 land users) remains computationally expensive in both time and cost.

Also, in addition to the graphs, I would be particularly interested in seeing (for example) a time-ordered list of bullet points that are output by the various institutions. (This is to get more of a flavor of what messages are being passed between agents.)

Thank you for your suggestion. We agree that a structured, time-ordered summary of institutional messages would help illustrate agent interactions. However, given the length and verbosity of the original textual outputs (which are already in bullet point format) condensing the outputs into a readable list without losing key details is challenging. For a more comprehensive view, we would suggest readers refer to the uploaded dataset, which contains the complete textual outputs. That said, we are open to exploring ways to present these results more effectively. We appreciate your feedback and will reflect on how best to integrate this in the revision.

---

## Referee Report (RR1)

Dear Zeng et al.,

Thank you very much for addressing my comments, feedback and concerns in such great detail. Thank you also for providing such detailed explanations in response to my questions.

It was a pleasure to read your revised manuscript. I found that the manuscript has greatly improved in clarity, structure and language. The terminology used is much more consistent and the definitions you provided for all terms make it much easier to understand the methodology and results. In my opinion, your efforts have assured that researchers from many different fields can understand and appreciate your work. I believe the revised manuscript has improved to such an extent that it is ready for publication.

Below you can find just some very minor comments. I refrained from providing detailed comments again because I believe the quality of the paper to be high enough for publication and do not wish to delay the process unnecessarily. Potentially, it would be good to read through the paper one last time in detail for spelling and grammar mistakes (although I did not encounter any when reading the paper).

**I have some final very minor comments:**
- *Table with definitions in the response letter (page 35)*: I was very happy to find a very clear table with an overview of terms and definitions in the context of the research. It helped me greatly to interpret the methodology, results and discussion of the paper. However, I did not find this table in the main paper (potentially this is by accident?). Could you please include the table you created in the main text.
- Thank you very much for explaining that the experiments mainly serve "proof-of-concept and exploratory purposes" in Section 2.3 of the methodology. This was unclear to me the first time I read the paper and made me read the paper in a very different way.
    - *Recommendation 1 – last paragraph introduction*: I would recommend to stress the purpose of the experiment (described in Sentence 3-6 of Section 2.3) in the last paragraph of the introduction. I notice you do use words like "explore", which I now understand refers to what you describe in Section 2.3. However, it may be good for the reader to realize the full purpose a little bit earlier, so that they do not read the paper with too high expectations.
    - *Recommendation 2 – abstract*: I am wondering if you could slightly revise sentence 21 ("Illustrative numerical experiments…") of the abstract to reflect on the purpose in the way you do in Section 2.3. I realize you have likely added the word "illustrative" to hint towards this. However, I am wondering if you could use the words "exploratory purposes" or "proof-of-concept" you use in Section 2.3 instead, as this fully clarified the purpose to me.
- *Robustness (Section 4.3)*: Thank you for defining robustness in detail and clarifying what you mean by this in the discussion of the paper. Would it be possible to reflect on robustness in the methodology as well (e.g., at the end of the methodology), so

that the reader knows a reflection on robustness is coming and what you mean by robustness?

- *Sentence 772 ("In such cases, a robust and...")*: Would it be possible to expand on what a well-designed error-handling mechanism could entail?

---

## Author Response (AR2)

**Combined Reply on RC1 and RC2**

**Response to Editor**

Dear Dr. Zeng,

Thank you very much for the detailed response to the reviewers' comments. Both were full of appraisal of your work in response to their comments. Both have some minor changes to add, but this should be easy to implement quickly. I especially support the request by rev#1 to include the table with definitions in the main text. As I am convinced that a response to reviewers should always not only lead to an explanation to the reviewer but also to future readers who may have similar questions.

Cheers
Christoph

Dear Editor Müller,

Thank you very much for your feedback and for the constructive suggestions from the anonymous reviewers.

We totally agree that including a table of definitions in the paper can help readers understand the research, which was our initial attempt during the Discussion phase. However, we did not do so in the previous version of the manuscript. After careful consideration, we have decided to put the definition table in the appendix. We have explained the rationale of our revision in response to Reviewer #1 (on the next page).

We appreciate your assistance throughout the review process.

Sincerely,
Yongchao Zeng
* * *
**Response to Reviewer #1**

Dear Zeng et al.,

Thank you very much for addressing my comments, feedback and concerns in such great detail. Thank you also for providing such detailed explanations in response to my questions.

It was a pleasure to read your revised manuscript. I found that the manuscript has greatly improved in clarity, structure and language. The terminology used is much more consistent and the definitions you provided for all terms make it much easier to understand the methodology and results. In my opinion, your efforts have assured that researchers from many different fields can understand and appreciate your work. I believe the revised manuscript has improved to such an extent that it is ready for publication.

Below you can find just some very minor comments. I refrained from providing detailed comments again because I believe the quality of the paper to be high enough for publication and do not wish to

delay the process unnecessarily. Potentially, it would be good to read through the paper one last time in detail for spelling and grammar mistakes (although I did not encounter any when reading the paper).

Thank you very much for your kind comments. As per your suggestion, we re-read the manuscript to identify and correct any remaining grammatical or spelling issues.

**I have some final very minor comments:**

*Table with definitions in the response letter (page 35)*: I was very happy to find a very clear table with an overview of terms and definitions in the context of the research. It helped me greatly to interpret the methodology, results and discussion of the paper. However, I did not find this table in the main paper (potentially this is by accident?). Could you please include the table you created in the main text.

Thank you for the suggestion.

During the Discussion phase, we initially considered including the definition table in the main manuscript. However, during the actual revision process, we found it was verbose to say, e.g., "see Table x for definition" repeatedly to guide the reader's attention. It is also inconvenient for readers to frequently shift back and forth between the main text and the table. Therefore, instead of relying on a separate table, in the previous revision, we chose to integrate the definitions/explanations within the main text at points where they fit best in terms of context coherence and text fluency.

That said, we remain open to including the table in the manuscript. We have placed the table in Appendix A and added a sentence to the last paragraph of Introduction: "To aid in interpretation, the definitions and explanations of key terminologies used throughout this paper are summarized in Table A1."

This revision may allow readers to engage with the main text without disruption while still easily locating specific definitions when needed.

Thank you very much for explaining that the experiments mainly serve "proof-of-concept and exploratory purposes" in Section 2.3 of the methodology. This was unclear to me the first time I read the paper and made me read the paper in a very different way.

*Recommendation 1 – last paragraph introduction*: I would recommend to stress the purpose of the experiment (described in Sentence 3-6 of Section 2.3) in the last paragraph of the introduction. I notice you do use words like "explore", which I now understand refers to what you describe in Section 2.3. However, it may be good for the reader to realize the full purpose a little bit earlier, so that they do not read the paper with too high expectations.

Done. We have now moved the sentences emphasizing the research purposes from Section 2.3 to the last paragraph of Introduction. Please see the manuscript with tracked changes.

*Recommendation 2 – abstract*: I am wondering if you could slightly revise sentence 21 ("Illustrative numerical experiments…") of the abstract to reflect on the purpose in the way you do in Section 2.3. I realize you have likely added the word "illustrative" to hint towards this. However, I am wondering if you could use the words "exploratory purposes" or "proof-of-concept" you use in Section 2.3 instead, as this fully clarified the purpose to me.

Good suggestion. We have added "exploratory purposes" to Abstract and revised the relevant sentences accordingly.

*Robustness (Section 4.3):* Thank you for defining robustness in detail and clarifying what you mean by this in the discussion of the paper. Would it be possible to reflect on robustness in the methodology as well (e.g., at the end of the methodology), so that the reader knows a reflection on robustness is coming and what you mean by robustness?

Done. Please see the last paragraph in Section 2.4.1. We have also accordingly removed the first three sentences in 4.3 to avoid repetition.

Sentence 772 ("In such cases, a robust and…"): Would it be possible to expand on what a well-designed error-handling mechanism could entail?

Thank you for the suggestion. Please see the last paragraph of Section 4.2 for the added content.
* * *
**Response to Reviewer #2**

My suggestions for minor revisions have been well-taken care of. Thank you! I only note that there are still 2 more occurrences of the typo 'wiring' instead of 'writing' in Table 1. I guess those can also be corrected?

Thank you very much for pointing out these typos. We have now corrected them and examined the paper throughout in case of similar issues. Please see the corrections in Table 1 in the manuscript with tracked changes.